# Measurements of CFC-11, CFC-12, and HCFC-22 total columns in the atmosphere at the St. Petersburg site in 2009-2019

Alexander Polyakov, Anatoly Poberovsky, Maria Makarova, Yana Virolainen, and Yuri Timofeyev

St. Petersburg State University, 7–9 Universitetskaya Emb., St. Petersburg 199034, Russia

**Correspondence:** Alexander Polyakov (a.v.polyakov@spbu.ru)

**Abstract.** Monitoring of atmospheric anthropogenic halocarbons plays an important role in estimating the ozone layer recovery and controlling the implementation of international agreements on emissions of ozone depleting substances. Within the NDACC network, regular FTIR measurements can provide information on abundancies of halocarbons on a global scale. We improved retrieval strategies for deriving the CFC-11 ($CCl_3F$), CFC-12 ($CCl_2F_2$), and HCFC-22 ($CHClF_2$) atmospheric
5 columns from IR solar radiation spectra measured by the Bruker IFS125HR spectrometer at the St.Petersburg site (Russia). We used the Tikhonov–Phillips regularization approach for solving the inverse problem and suggested the optimized values of regularization parameters. We tested the strategies developed by comparison of the FTIR measurements with independent data. The analysis of time series of column-averaged dry-air mole fractions (Xgas) measured in 2009–2019 gave the mean values of $225\,\text{pptv}$ (CFC-11), $493\,\text{pptv}$ (CFC-12), and $238\,\text{pptv}$ (HCFC-22). Trend values total $-0.40\,\%\,\text{yr}^{-1}$ (CFC-11), $-0.49\,\%\,\text{yr}^{-1}$
10 (CFC-12), and $2.12\,\%\,\text{yr}^{-1}$ (HCFC-22).

We compared the means, trends and seasonal variability of $X_{CFC-11}$, $X_{CFC-12}$, and $X_{HCFC-22}$ to that of 1) near–ground volume mixing ratios (VMRs) measured at the observational site Mace Head, Ireland (GVMR); 2) the mean in the 8–12 km layer VMRs measured by ACE-FTS and averaged over 55-65°N latitudes (SVMR); 3) Xgas values of the Whole Atmosphere Community Climate Model for the St. Petersburg site (WXgas).

15 In general, the comparison of the Xgas with the independent data showed a good agreement of their means within the systematic error of measurements considered. The trends observed over the St. Petersburg site demonstrated the smaller decrease rates for $X_{CFC-11}$ and $X_{CFC-12}$ than that of the independent data, and the same increase rate for $X_{HCFC-22}$. In a whole, Xgas, SVMR, and WXgas showed qualitative similar seasonal variations, the GVMR variability was significantly less; and only the $WX_{HCFC-22}$ variations were essentially smaller than that of $X_{HCFC-22}$ and $SVMR_{HCFC-22}$. The retrieval tech-
20 niques presented may be useful in further development of unified strategies for halocarbons retrieval at all sites of the NDACC observational network that use the Bruker IFS125HR spectrometers for IR solar spectra measurements.

# 1 Introduction

Since the middle of the 20th century, anthropogenic trace gases, the molecules of which contain halogens, due to their specific physical and chemical properties have been actively used in climatic and refrigeration industry as well as in various propellants. Molina and Rowland (1974) have shown that these gases play an important role in the depletion of the stratospheric ozone. In particular, the photolysis of $CCl_3F$ (trichlorofluoromethane, CFC-11) and $CCl_2F_2$ (dichlorodifluoromethane, CFC-12) in the stratosphere leads to the appearance of the active chlorine which is involved in ozone depletion reactions. The WMO (2018, Appendix A) estimates the ozone depletion potential (ODP) of CFC-12 as $0.73 - 0.81$ (ODP of CFC-11, chosen as a reference, equals 1). Although the major content of these gases is concentrated in the troposphere, in the equatorial region the global circulation moves them out into the lower and middle stratosphere and transports to high–latitude regions. In the stratosphere, CFCs are photochemically decomposed to chlorinated free radicals (Cl, ClO) that are deactivated into chlorine reservoirs HCl, $ClONO_2$, and HOCl (WMO, 1985, Chapter 3). In polar regions, the heterogeneous reactions on the surfaces of polar stratospheric clouds and cold sulfate aerosols convert inert reservoir molecules into active forms that photolyze producing free radicals and cause the chemical ozone depletion in spring through catalytic cycles resulting up to the appearance of so–called ozone holes (Solomon et al., 2014).

As the result of the Montreal Protocol and its amendments and adjustments that restricted the production of chlorofluorocarbons (CFCs) (see (WMO, 2018)), the industry moved away from CFCs to less ozone–depleting hydrochlorofluorocarbons (HCFCs), especially $CHClF_2$ (chlorodifluoromethane HCFC-22). Although the ODP of HCFC-22 is much lower than that of CFCs, it is an ozone-depleting substance, too. Ozone depletion by HCFC-22 is primarily associated with the heating of the stratosphere, and its ODP, although small, totals 0.024–0.34.

CFC-11 and CFC-12, like HCFC-22, also absorb the infrared radiation, therefore they are all greenhouse gases. The Global Warming Potential (GWP) represents the integrated radiative forcing (RF) for a conditional time horizon ($20, 100, 500$ years) caused by the emission of a unit mass of a gas relative to the same RF value of $CO_2$ that is chosen as a reference for estimating the GWP of other gases. According to WMO (2018, Appendix A), the GWPs for 100 years are 5160 for CFC-11, 10300 for CFC-12, and 1780 for HCFC-22. One of the reasons of the high values of the GWP of these gases is their long lifetimes: $52, 102, 11.9$ years, respectively. Due to their long lifetimes, these gases are also good indicators for studying the transport and mixing processes in the upper troposphere and lower stratosphere (e.g. Hoffmann and Riese, 2004).

After Molina and Rowland (1974) reported that CFCs accumulated in the Earth's atmosphere led to an increased rate of ozone depletion, the attention of both scientists and policymakers to the ozone hole problem had been increased. Nowadays, monitoring of the ozone and other stratospheric gases as well as the ozone depleting substances including CFCs are crucial for testing the theories of the ozone hole formation mechanism (Cracknell and Varotsos, 2009).

The Montreal protocol from 1987, which came into force in 1989, limited production and consumption of CFCs. Later on, in 1992 in Copenhagen and in 1995 in Vienna, phase-out of CFCs was stated by the end of 1995 in developed countries and by the end of 2010 in developing countries. Therefore, the atmospheric burden of CFC-11 and CFC-12 was declining at an average rate of $0.7$–$1.2$ and $0.4$–$0.5$ % per year, respectively (Brown et al., 2011). ACE-FTS satellite measurements in last 16 years

(Bernath et al., 2020) illustrated the success of the Montreal Protocol by estimated decreasing trends in CFC-11 ($-0.53$ % per year) and CFC-12 ($-0.61$ % per year) abundancies and a slowing rate of increase in HCFC-22 abundancies (1.8 % per year). ACE-FTS estimates are made for ranges of 60 S to 60 N in latitude and of 5.5 km to 10.5 km in altitude. Nevertheless, having been accumulated in the troposphere, CFC-11 still provides a quarter of all chlorine reaching the stratosphere. The time needed for recovery of the ozone layer among other factors depends on the sustainability of the reduction in the concentration of CFC-11, CFC-12 and other halocarbons in the atmosphere.

Based on the 2015–2017 data, Montzka et al. (2018) showed that the rate of change in the CFC-11 atmospheric concentration decreased by approximately half to $-0.4\,\%\,\mathrm{yr}^{-1}$, assuming that this slowdown is caused by the emergence of new unregistered sources. This finding enhances the importance of monitoring the atmospheric concentration of CFC-11. The maximum of the CFC-12 atmospheric concentration was observed in the early 2000s, since then its steady decrease has been detected with an average rate of 0.4–0.5 % per year (AGAGE network, http://agage.mit.edu/data/agage-data). As HCFCs are the 'transitional substances' for the replacement of CFCs, their production has increased rapidly in developed countries in the 1990s and peaked in the mid–1990s. Under the Montreal Amendment (1997) all countries must gradually phase down HCFCs. In September 2007 it was decided to accelerate the phase-out of HCFCs. Developed countries had been reducing their consumption of HCFCs and completely phasing them out by 2020. Developing countries agreed to start their phase out process in 2013 and are now following a stepwise reduction until the complete phase-out of HCFCs by 2030.

Until recently, two data sources were mainly used to study the trends and seasonal variations of the target gases: local measurements of near ground concentrations (for example, the AGAGE networks (Dunse et al., 2005), and NOAA's Halocarbons & other Atmospheric Trace Species Group NOAA CAMP (Montzka et al., 1993)) and satellite limb measurement experiments ILAS, ACE–FTS, MIPAS (Hoffmann et al., 2008; Mahieu et al., 2008; Eckert et al., 2016; Kellmann, et al., 2012; Boone et al., 2020). In contrast to satellite and in situ measurements near ground, ground–based Fourier Transform Infrared (FTIR) measurements of the solar radiation are sensitive to the total columns (TCs) of atmospheric gases down to the Earth's surface. The FTIR method complements the information obtained by the first two methods, although it does not allow the detailed information on the vertical gas distribution to be retrieved.

First FTIR measurements of atmospheric HCFC-22 were performed from the balloon in early 80-s (Goldman et al., 1981). Spectral resolution of these measurements did not allow halocarbons to be measured from the surface. Later, with the appearance of high-resolution instruments, halocarbons started to be derived with ground-based FTIR spectrometers. In last decades, TCs of halocarbons are measured by ground-based FTIR method more actively (e.g. Notholt, 1994; Rinsland et al., 2005, 2010; Zander et al., 2005; Mahieu et al., 2010, 2013, 2017; Zhou et al., 2016; Prignon et al., 2019).

Time series of CFC-11, CFC-12, and HCFC-22 TCs above Jungfraujoch station, Switzerland, are presented in periodic WMO reports on Scientific Assessment of Ozone Depletion (e.g. WMO, 2018).

Within the NDACC network (http://www.ndsc.ncep.noaa.gov), regular FTIR measurements provide the information on the TCs of a number of atmospheric trace gases, including halocarbons, with a large spatial coverage (at 19 out of 77 network stations located at latitudes between 78° S to 80° N). Mahieu et al. (2017) reported on the results of R-142b measurements along with the comparison with independent data and the trend estimates. Zhou et al. (2016) showed the results of CFC-11,

CFC-12, and HCFC-22 measurements at two NDACC sites on Réunion Island for the period of 2004–2016, including the trend estimates and the comparison with the satellite data. Prignon et al. (2019) proposed a technique for estimating two partial columns and TCs of HCFC-22 at the Jungfraujoch mountain station and corresponding time series of HCFC-22 TCs for 1988-2017 along with the trend analysis for various time periods.

The archive of ground–based spectroscopic measurements of IR solar radiation performed at the NDACC site St. Petersburg (Timofeyev et al., 2016; Virolainen et al., 2017) since 2009 can be used to derive the data on TCs of CFC-11, CFC-12, and HCFC-22. First in Russia estimates of the CFC-11 TCs using the FTIR method and the original retrieval technique were given by Yagovkina et al. (2011). Polyakov et al. (2018) presented the preliminary results of the CFC-11, CFC-12, and HCFC-22 TCs retrieval for the period of 2009-2016 using the SFIT4 software, version 0.9.4.4, described by Hase et al. (2004). It should be noted that the SFIT4 software is a versatile tool, and it is necessary to customize it for a specific task through a selection and tuning of numerous parameters. Polyakov et al. (2018) selected these parameters based on the studies at other NDACC sites (Mahieu et al., 2010; Zhou et al., 2016) and the general recommendations of the IR working group (IRWG) of the NDACC network. However, these first results raised a number of questions, in particular, an unreasonably large scatter of the TCs values and significant seasonal variations. Later study showed that the observed scatter and seasonal variability were not due to objective reasons, but to peculiarities of the processing retrieval technique.

The informativeness of the FTIR spectra with respect to target gases abundancies is low due to several reasons. First, the absorption of target gases is small. Therefore, for example, even for low sun (solar elevation of about 15°), the transmission of solar radiation due to CFC-11 absorption is greater than 90 %, due to HCFC-22 absorption is close to 75 %, and only for CFC-12, the transmission is close to 50 %. Secondly, there is an absorption of interfering gases in the spectral range considered. Thus, the CFC-11 absorption band is overlapped with several strong water vapor absorption lines and $HNO_3$ absorption band, each of the CFC-12 and HCFC-22 absorption bands is overlapped with a wing of water vapor absorption line. Finally, the CFC-12 and at a larger extent the CFC-11 bands have smoothed spectral dependency of absorption that requires to use wide micro-windows for retrieving their abundancies, $2\,\mathrm{cm}^{-1}$ for CFC-12 and not smaller than $30\,\mathrm{cm}^{-1}$ for CFC-11. These factors cause the difficulties in halocarbons retrieval from FTIR spectra measurements.

Later on, the retrieval techniques for estimating the CFC-11, CFC-12, and HCFC-22 TCs by the FTIR method at the St. Petersburg site were refined and improved. These techniques were described in detail by Polyakov et al. (2019a, b, 2020a). In the current study, we used the Tikhonov–Phillips (T–Ph) approach to improve the retrieval strategy as in contrast to the optimal estimation (OE) method it does not "pull" the solution to the apriori profile for measurements with low information content. Moreover, the T-Ph approach allows more stable results to be derived than the OE method (e.g. Senten et al., 2012). We presented the main features of the techniques developed and analyzed the using of the T–Ph approach. The time series of CFC-11, CFC-12, and HCFC-22 TCs were extended until the fall of 2019. The time series of the TCs were analyzed and compared to the independent measurements and numerical modeling data.

## 2 Technique for inverting the spectroscopic measurements

### 2.1 Spectroscopic measurements

The main features of the ground–based station, observational system and the technique for measuring the solar spectra used in this study were described in detail by Timofeyev et al. (2016).

The St. Petersburg site is located in Peterhof, $30\,\mathrm{km}$ west of St. Petersburg city. The latitude of the site $59.88°\,\mathrm{N}$ predetermines winter measurements with a low Sun elevation: in December–January, the maximum height of the Sun does not usually exceed $20°$; spectroscopic measurements are performed up to the Sun's height of $5°$. Due to peculiarities of the local weather, measurements are mainly ($76\,\%$) carried out in spring and summer seasons. The spectra analyzed are obtained without any additional apodization of the interferograms, their spectral resolution is $0.005\,\mathrm{cm^{-1}}$. The observational system is based on a Bruker IFS125HR Fourier spectrometer, but some of the equipment is non–standard. In particular, before February 2016 a non–standard (for the IRWG-NDACC sites) spectral filter (hereinafter F3) was used for measurements in the spectral region with target gases absorption bands. Since this filter was plane–parallel, a parasitic interference arose in it, leading to the appearance of an effect of the optical resonance ("channeling"), see (Blumenstock et al., 2020). Moreover, a home-made solar tracking system is used.

The period of the channeling is caused by the material and the thickness of the filter, and in the spectral region $800 - 900\,\mathrm{cm^{-1}}$ it is close to $1.1\,\mathrm{cm^{-1}}$, while the amplitude of the channeling varies from zero to a few percent value depending on random filter positioning parameters. To analyze the presence and the amplitude of the channeling, we performed the Fourier analysis in the most transparent spectral range $892 - 905\,\mathrm{cm^{-1}}$ for harmonic components with periods in the intervals of $1 - 1.25\,\mathrm{cm^{-1}}$. The channeling amplitude was calculated relative to the mean signal value in this spectral range.

It is worth mentioning that the SFIT4 software supports the accounting for channeling and its compensation in a spectrum. However, our estimations showed that when the channeling amplitude exceeded $2\,\%$, this compensation became unsatisfactory due to a significant increase of scatter in the results of retrievals. Thus, we excluded such spectra from the further processing. In addition, we analyzed the autocorrelation coefficient of a dark noise in the range of $660 - 680\,\mathrm{cm^{-1}}$ (except a slope) and excluded the spectra with the averaged autocorrelation coefficient greater than $0.1$. A large autocorrelation coefficient of the dark noise indicates the presence of external influences on a measurement process. Moreover, we excluded from the further processing the spectra that were measured at the time when a haze or cloudiness was observed in any part of the sky, since the use of these spectra also noticeably increases the scatter of the results.

As a result of the described filtering, 2901 from 3523 (i.e. 82 %) spectra measured before February 2016 were selected for further processing. In February 2016, the F3 filter was replaced by the standard IRWG NDACC filter f6 which wedge–shaped design eliminates the channeling. Thus, the quality of measurements was improved, and 1903 from 1958 spectra were selected, giving in a sum 4804 spectra for 2009-2019 period.

## 2.2 Main parameters of the retrieval technique

In previous studies, Polyakov et al. (2019a, b, 2020a) determined a number of parameters of the retrieval strategy using the SFIT4 code for deriving the TCs of the target gases from the FTIR measurements at the St. Petersburg site: the boundaries of microwindows, the mean/apriori profiles of the measured gases, the magnitude and variability of the zero level, periods for taking into account (or excluding) the channeling, and the background shape of a spectrum (BSS).

The criteria used for optimization of retrieval parameters are briefly described below. As we mentioned above, the lifetime of the target gases in the atmosphere is more than 10 years. In addition, CFC-11 and CFC-12 have no known active sources of emission. Therefore, we expect the stability of their retrieved columns during both the each day and the whole period of measurements (excluding trend). At a lesser extent due to its continuous production, the same criterion is valid for HCFC-22, at least for intraday variability. Thus Polyakov et al. (2019a, b, 2020a) used the stability of the retrieved total columns in terms of minimal root-mean-squared (RMS) SD of the TCs for all days of measurements as the main criterion in selecting the retrieval parameters. Another important retrieval parameter is the number of degrees of freedom for signal (DFS) (Rodgers, 2000, p. 19) for target gases. As a criterion for optimization, the SD of DFS was minimized. The estimate of total systematic and random measurements errors was also considered. Finally, the residuals (differences between spectra measured and calculated with the retrieved atmospheric state), for estimating of which the RMS residuals normalized to the unit, calculated in the SFIT4 software, and denoted as $\chi^2$, were analyzed. It should be noted that without additional analysis, the listed criteria do not unambiguously determine the optimality of the retrieval technique. The estimate of the TCs measurement errors can be considered as one of the main criteria, but the adequacy of the measurement model also should be taken into account. Thus, for example, by including during spectra analysis the additional unknown parameter as channeling, we increase the measurements errors, however, if we exclude it and residuals get larger, it will indicate that the parameters used are inadequate for real measurements, i. e. the actual presence of channeling in the spectra. Table 1 presents the main optimized parameters obtained in previous studies.

**Table 1.** The main parameters of the inversion of the spectra for deriving the TCs of halocarbons obtained by (Polyakov et al., 2019a, b, 2020a).

| Gas | Microwindow, $cm^{-1}$ | Other gases | $H_2O$ spectroscopy | Accounting for channeling (beam) |
|---|---|---|---|---|
| CFC-11 | 830–860 | $H_2O$(profile), $CO_2$, $O_3$, $HNO_3$, $COCl_2$ (columns) | HITRAN 2016 | $1.12\,cm^{-1}$ before 2016 |
| CFC-12 | 1160–1162 | $H_2O$, $O_3$, $N_2O$, $CH_4$ (columns) | HITRAN 2009 | $1.26\,cm^{-1}$ before 2016 |
| HCFC-22 | 828.75–829.4 | $CO_2$, $O_3$, $H_2O$ (columns) | HITRAN 2009 | $1.1\,cm^{-1}$ before 2016 |

While processing the measured spectra, the spectroscopic parameters supplied as a part of the SFIT-4 software are used. Target gas and $COCl_2$ absorption is calculated based on pseudo–lines (see mark4sun.jpl.nasa.gov/pseudo.html for pseudo–lines), other interfering gases absorption is calculated based on spectroscopic information from the HITRAN database which is described in detail by Polyakov et al. (2019a, b, 2020a). The apriori information on the physical state of the atmosphere is taken

from the NCEP CPC (presented on ftp.cpc.ncep.noaa.gov/ndacc/ncep); water vapor profiles used in the retrieval are independently derived from the FTIR measurements, the technique is described by Virolainen et al. (2017). The apriori profiles of other interfering gases are taken from the Whole Atmosphere Community Climate Model (WACCM) (Park et al., 2013). As a first guess for target gases, the mean profiles of the WACCM dataset for the 2009–2019 period are used. The using of a wide spectral window for CFC-11 retrieval ($30\,\mathrm{cm}^{-1}$, see Table 1) is unusual for deriving the information on the gas content from the high resolution IR spectra and requires the non-standard approach for considering the BSS. This approach was described in detail by Polyakov et al. (2020a); the main features of this approach are listed below.

The constant and important factor that determines the BSS is the filter spectral transmission function (STF). We have measured the STF in a special experiment using an artificial light source.

Repeated measurements of the STF showed that over time they exhibit a specific spectrum of absorption by amorphous water ice (AWI) formed on the HgCdTe detector at the temperatures that has a detectorcooled by liquid nitrogen (e.g. Hudgins, et al., 1993; Lynch, 2006). The absorption of radiation by AWI depends on its thickness which increases during the measurement period and decreases during the period of inactivity of the instrument when the receiver is not cooled. In addition, the water vapor from the atmospheric air gradually (on a monthly scale) seeps into the evacuated zone of the instrument and also leads to an increase of the AWI thickness. To compensate for its variability, the BSS was refined with the second–degree wavenumber polynomial implemented in the SFIT4 code. With turning on one more variable, the correction of the BSS curvature specified by the coefficient at the second power of the wavenumber (hereinafter – curvature value) can lead to "overfreedom" of the solution. To avoid this, the apriori curvature value uncertainty was limited. The parameters for compensating the BSS due to absorption by the AWI were selected in two steps. We minimized the intraday variability of the CFC-11 TCs in a series of spectra processing and, on the first step, got the apriori thickness of the AWI ($0.3\,\mu\mathrm{m}$ for F3, $0.9\,\mu\mathrm{m}$ for f6 filter) with the apriori curvature value of 0. On the next step, we optimized the value of apriori curvature uncertainty as $10^{-6}$ for both filters.

Water vapor continuum makes a significant contribution to radiation attenuation by the atmospheric water vapor (Mlawer et al., 2012). Our calculations have shown that radiation absorption by water vapor continuum in the considered spectral region under conditions of the St. Petersburg site can significantly exceed $50\,\%$. For a $30\,\mathrm{cm}^{-1}$ window, the selectivity of continual uptake is sufficient to influence the spectra processing results. To calculate the water vapor continuum, we used a free–distributed computer code (MT_CKD, 2017) and the daily profiles of water vapor independently derived from the FTIR measurements (Virolainen et al., 2017). In the first approximation, the contribution of the water vapor continuum to absorption is proportional to the water vapor partial pressure squared, and it can be detected only in a very humid atmosphere. We estimated the contribution of water vapor continuum numerically by analyzing spectra with and without it considering on the most humid days in 2018: July 29, August 2 and 9. The neglecting continuum absorption in these days led to overestimates of the CFC-11 TCs by an average of 2.9 %. Although this value is less than the measurement error (see Table 4), it systematically depends on the water vapor content. Therefore, it is necessary to consider the water vapor continuum. In Fig. A1 of Appendix A, we present an example of transmission due to absorption by water vapor continuum.

Thus for CFC-11 processing, we took into account STF, AWI variability and water vapor continuum.

## 2.3 Method for solving the inverse problem

In earlier studies, Polyakov et al. (2018) used the OE method (Rodgers, 2000, p. 65) for solving the inverse problem, i.e. apriori information was given in the form of a mean profile and a covariance matrix with 0.05 relative uncertainty at all altitudes and a correlation coefficient of 5 km built on the basis of the WACCM dataset for the period of 1980-2020. In accordance with WMO (2018), all three target gases are characterized by a long lifetime, therefore the low variability of their content on a time scale of less than several years is expected and confirmed, for example, by measurements of AGAGE and HATS networks (Dunse et al., 2005; Montzka et al., 1993). In this case, their long–term variability turns out to be predominant. Such a character of the variability, especially in combination with a small DFS (e.g. Zhou et al., 2016; Polyakov et al., 2019a, b, 2020a), is better described by the Tikhonov–Phillips approach presented by Tikhonov (1963) and Phillips (1962). Unlike the OE, the T–Ph approach does not "pull" the solution to the mean profile but only limits its vertical variability. The use of the first order T–Ph regularization (Tikhonov, 1963) in retrieving the TCs of trace gases is described in detail, for example, by Sussmann et al. (2011). Besides, when choosing the type of the regularisation (OE or T–Ph), it should be taken into account that the OE approach, with the apriori information given in a form of normal distribution of the target gas profile, requires the use of the covariance matrices for describing the variability of the target gases profiles, preferably the real covariance matrices, that are unavailable for halocarbons considered. Although there are datasets of available satellite measurements (ACE-FTS and MIPAS) of halocarbons, they cannot be used for constructing covariance matrices for several reasons: 1) The data of limb or solar occultation usually do not provide information on atmospheric gases below $7 - 8 \, \mathrm{km}$, whereas the target gases are mainly located in the troposphere. The use of extrapolation to construct covariance matrices to altitudes with the largest content of a gas makes it meaningless to use the measured profiles. 2) The horizontal resolution of these measurements is of hundreds (typically $300 - 500$) km, thus satellite measurements are in a poor agreement with local vertical profiles, for which the covariance matrix is needed. 3) The construction of local matrices for St. Petersburg is complicated due to a small number of ACE-FTS measurements for this location and a short period of MIPAS measurements. Therefore we chose the T–Ph regularization for spectra processing.

For opimization of the regularization parameter $\alpha$, we used a technique based on minimizing the intraday variability of the TCs suggested by Sussmann et al. (2011). In addition, we analyzed the spectral residual RMS, sometimes denoted as $(\chi^2)$ , and the values of DFS. For analysis of the regularization parameter, we used spectroscopic measurements in 2017 which were characterized by a fairly stable quality of measurements, a low noise level, and a possibly more uniform distribution of measurements throughout the year including the winter months. Additionally, the year 2017 was chosen due to the measurements with the f6 filter which is currently used by other sites of the IRWG NDACC network. The year 2015 was chosen for testing the F3 filter; calculations confirmed that the values of $\alpha$ obtained for 2017 for f6 filter are also optimal for F3 filter.

Figure 1 depicts the RMS intraday variability of the TCs of target gases as a function of $\alpha$ for 2017. It is worth mentioning that the difference between the curves for different halocarbons in Fig. 1 is meaningful. The presence of a pronounced minimum for CFC-12 is due to a larger information content of spectral measurements with respect to CFC-12 abundancies compared to CFC-11 and HCFC-22 (DFS for CFC-12 is 1.2, CFC-11 - 1.05, HCFC-22 - 1.0, see Table 3). The reason for this is a weak

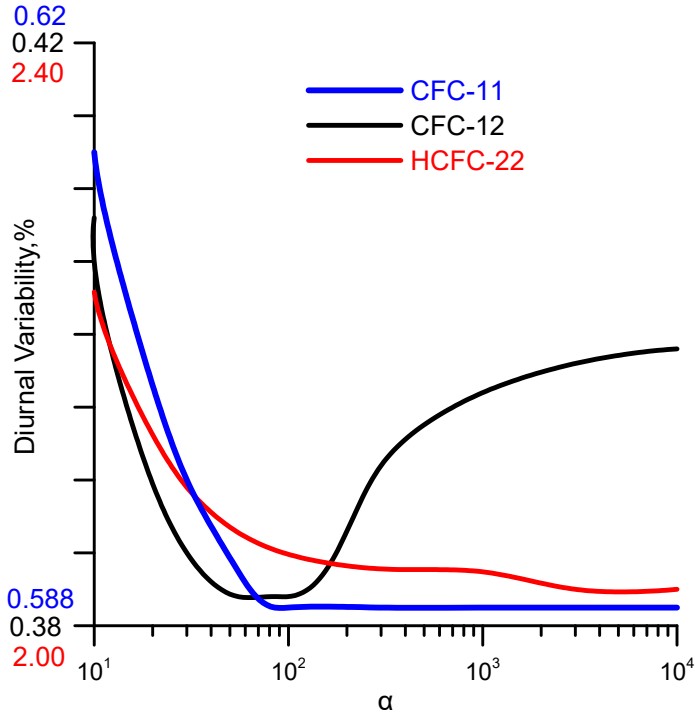

**Figure 1.** Dependence of the TCs intraday variability on the regularization parameter $\alpha$.

absorption of interfering gases in the spectral range for CFC-12 retrievals. Thus, for CFC-12, an increase of the regularization parameter, which tightens the requirement for spectrum smoothness, leads to the suppression of useful information on the elements of the vertical gas distribution which is contained in a spectrum. Consequently, the intraday variability of retrievals is increasing. For CFC-11 and HCFC-22, the informativeness of spectral measurements is less, DFS is close to 1, therefore, large values of the parameter $\alpha$ and the corresponding requirements for smoothness do not contradict the information contained in a spectrum.

For CFC-11, the minimum of the intraday variability of $0.589\,\%$ is reached asymptotically for all values of $\alpha$ not less than 85, and the DFS at $\alpha = 85$ differ from 1 (DFS= 1.08). For CFC-12, the optimal value of the regularization parameter $\alpha = 85$, this value corresponds to the intraday SD minimum of $0.382\,\%$, and DFS totals 1.18. For HCFC-22, the minimum of intraday variability of $0.398\,\%$ is reached asymptotically for all values of $\alpha$, starting from $3 \times 10^3$, the DFS for all these values amounts to 1.00, and both parameters do not change for $\alpha$ greater than $3 \times 10^3$. This can be interpreted as the complete absence of the information on the vertical profile of HCFC-22 in spectral measurements, i.e. only the information on the first guess profile multiplier (profile scaling approach). We performed the retrieval for both the profile scaling and the T–Ph approach with $\alpha = 3 \times 10^3$. It turns out that although the SFIT4 software gives practically the same results, the option with the profile scaling retrieval does not allow calculating the error budget. Therefore, we used the T–Ph method for HCFC-22 retrieval with $\alpha = 3 \times 10^3$.

It should be noted that the target parameter of the retrieval, TC, is calculated from the initially retrieved vertical profile of the gas. Therefore, it is important to control the retrieval of trace gases profiles. For all three target gases, figures 2–4 depict the sensitivity functions of the TCs to relative variations in the gases profile at different heights (left) (see about averaging kernel (AK) area (Rodgers and Connor, 2003, section 2.1)) and the examples of initial (first guess) and retrieved volume mixing ratio (VMR) profiles (right). All curves are shown for two typical measurements: in a fall–winter season with a low Sun elevation and a low humidity, and in summer, with a high for the site latitude Sun elevation and a wet atmosphere. All parameters in the figures are given for a regularization of both the OE and the T–Ph methods. Although the T-Ph regularization parameter was optimized, the covariance matrices of the OE method were taken from (Polyakov et al., 2018), where they were selected from general considerations.

Figures 2–4 demonstrate that the sensitivity, which for the ideal case should be equal to 1 at all heights, turns out to be noticeably lower (from 0.5 to 0.8 for different gases, seasons and methods) at the surface. Then sensitivities increase, reaching a maximum at heights of $8 - 12\,\mathrm{km}$ for CFC-11 and CFC-12, and at heights above $12\,\mathrm{km}$ for HCFC-22 which is due to a higher stratospheric content of HCFC-22. Above, the sensitivity decreases which can no longer be significant due to the fall of the VMR of the target gases. As seen from Fig. 2–4, the measurement conditions have a significant effect on the sensitivity functions. In winter, when the Sun elevation is low corresponding to a thicker atmosphere in the solar beam path and low water vapor content, the information content of measurements is higher than in summer. For all three gases, the sensitivity is far from the unit at a greater extent in the lower troposphere in high humidity conditions in summer. Using T–Ph approach and choosing the regularization parameter based on minimizing the intraday variability of TCs, we obtained DFS = 1 for HCFC-22; the DFS value of other two gases is close to 1 (1.05 and 1.20, see Table 3). Prignon et al. (2019) reported the higher values of DFS (DFS

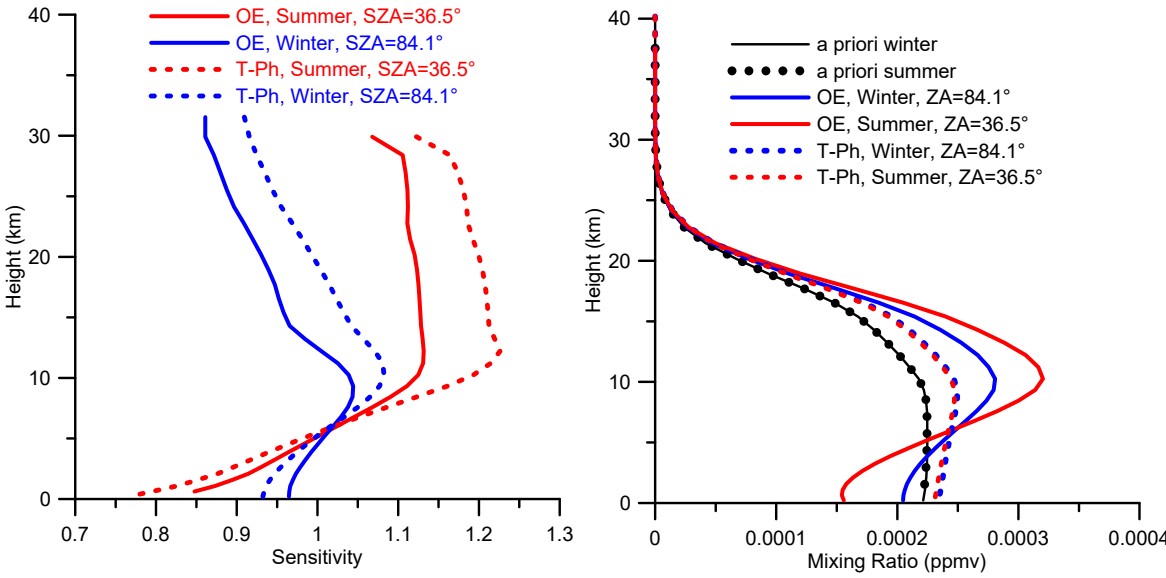

**Figure 2.** Sensitivity of CFC-11 TCs to variation of its VMR profile (left) and a priori and retrieved profiles (right) on 16 June 2018 09:54 (red) and 14 October 2018 13:57 (blue).

= 1.97) caused by T-Ph regularisation with parameter $\alpha = 9$ and a low atmospheric water vapor content above the mountain (3580 m a.s.l) site Jungfraujoch.

## 3   Results and analysis

The techniques described above were applied to processing the entire archive of spectral measurements at the NDACC site St. Petersburg for the period of 2009–2019.

### 3.1   The filtering of the results

Table 2 presents a number of statistical characteristics and the assessment of the total errors of the obtained TCs of the target gases.

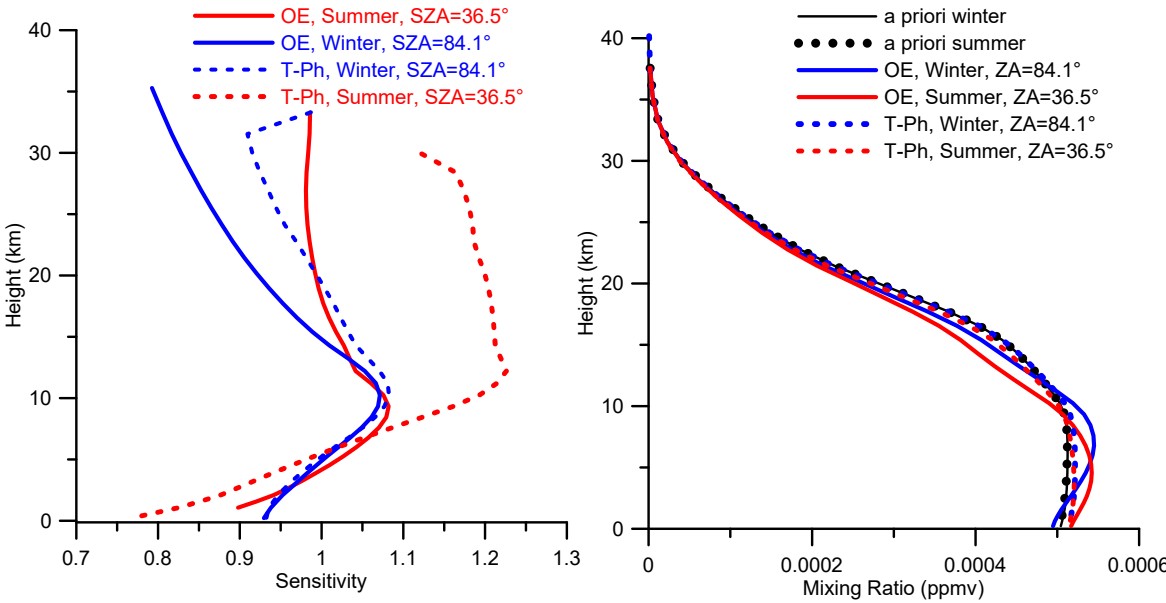

**Figure 3.** Sensitivity of CFC-12 TCs to variation of its VMR profile (left) and prior and retrieved profiles (right) on 16 June 2018 09:54 (red) and 14 October 2018 13:57 (blue).

**Table 2.** Summary of the statistics for retrieved halocarbons TCs before filtering. The numbers after the "±" sign indicate the standard deviation of the values.

| N | Parameter | CFC-11 | CFC-12 | HCFC-22 |
|---|---|---|---|---|
| 1 | Number of spectra/days | 4773/720 | 4768/718 | 4585/714 |
| 2 | RMS, % | $0.53 \pm 0.46$ | $0.45 \pm 0.55$ | $0.40 \pm 0.29$ |
| 3 | Total Systematic Error, % | $7.60 \pm 0.18$ | $2.26 \pm 0.16$ | $5.75 \pm 0.08$ |
| 4 | Total Random Error, % | $3.23 \pm 0.77$ | $2.56 \pm 0.94$ | $4.18 \pm 2.66$ |
| 5 | intraday SD, % | 1.35 | 0.70 | 5.63 |
| 6 | DFS | $1.07 \pm 0.09$ | $1.20 \pm 0.05$ | $1.00 \pm 0.00$ |

The first row in Table 2 shows the total number of spectra / days for which the TCs have been obtained. Although the total number of spectra taken for SFIT4 processing was 4804 (see section 2.1), we had to remove 31 spectra for CFC-11 as they

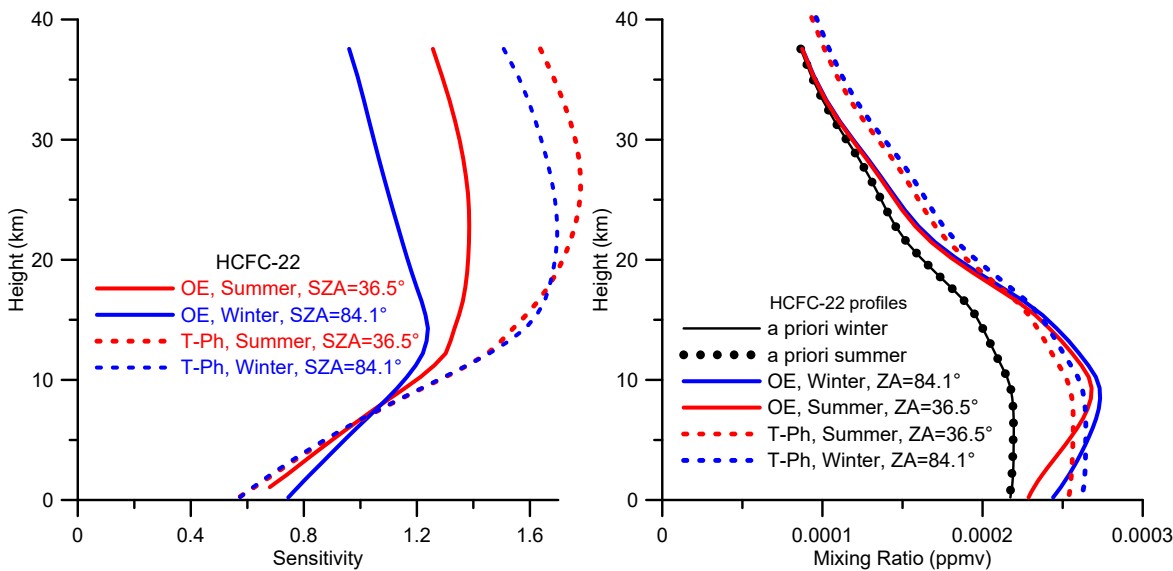

**Figure 4.** Sensitivity of HCFC-22 TC to variation of its VMR profile (left) and prior and retrieved profiles (right) on 16 June 2018 09:54 (red) and 14 October 2018 13:57 (blue).

were measured with incorrect filter (F3 instead f6 and vice versa). The retrieval technique for CFC-11 is very sensitive to the correctly defined filter (see section 2.2 on BSS for CFC-11). Thus the number of spectra for CFC-11 was 4773. The numbers in the Table 2 are different for different gases since the inverse problem solution algorithm implemented in SFIT4 does not always provide a solution. The total number of spectra suitable for processing for over more than 10 years of observations (from March 2009 to August 2019) is about 4500–4800, and they have been measured for about 720 days. Thus, on average, the FTIR measurements at the St. Petersburg site are carried out for 68 days per year. Such a relatively low annual frequency of the measurement days number is primarily due to the latitude and climatic features of the site.

As we observed some outliers in the HCFC-22 TCs time series before 2016, we discarded the TCs values that differ from the approximating line (trend) by more than 3 SD values. 219 measurements were excluded, thus HCFC-22 spectra number is less then spectra numbers for two other gases. On the next step we filtered the retrieved TCs using the following criteria: the deviation from the mean statistical characteristics presented in Table 2 should not be greater than 2×SD. Details of this selection are shown in Table B1 of the Appendix B.

Table 3 gives a summary of the measurement statistics of the retrieved TCs. The general information on the spectra analyzed is given in row 1 (the number of measurement days and single measurements) and in row 2 (the spectral residuals). The number of days is close to 670 and the number of the retrieved TCs is close to 3900 for each gas. The spectral residual is the most important parameter of the inverse task solving; it characterizes the quality of fitting the measured spectra by calculated one. Ideally, the spectral residual should be equal to the measurements noise level. For target gases, the mean values of the spectral residuals vary from $0.34$ to $0.52\,\%$ depending on the gas; it corresponds to the SNR values of $209, 280$, and $327$ for CFC-11, CFC-12, and HCFC-22, respectively. Since the spectrum in residual calculations is normalized to unit, SNR and residual there are reciprocal values, $\text{SNR} = 1/\text{residual}$ Comparing these values to the preliminary determined mean SNR in the opaque spectral range ($364, 351$ and $324$, in the same order of gases), we see that for CFC-11 and CFC-12 they are slightly less and for HCFC-22 are nearly the same. This means that for CFC-11 and CFC-12 the radiative transfer model and a set of parameters used, although satisfactory, but not ideally describe the absorption of radiation by the atmosphere and the observational system, whereas for HCFC-22 the retrieval technique works in the best way.

Rows 3–7 of Table 3 present the characteristics of the target gases retrievals, TCs and Xgas. Row 3 shows the means, and row 4 shows the RMS intraday variability of Xgas, which can be interpreted as their precision. Comparison of the RMS intraday variability values with the estimates of the random error (row 11 of Table 4) demonstrates that for HCFC-22 the intraday variability practically coincides with the total random error. The other two gases show a significantly different ratio, the intraday variability is noticeably less than the random error – $0.76$ vs $3.08\,\%$ for CFC-11 and $0.58$ vs $2.40\,\%$ for CFC-12. Therefore, the random error has a significant component of a systematic nature during one day of measurements, but randomly changes from one day to another. It should be noted that the temperature profile changes insignificantly during a day, so the intraday variability of the Xgas contains a related component and exceeds the contribution of a total random noise of spectroscopic measurements. Thus, the resulting error budget estimates and the intraday variability of the results are mutually consistent.

The DFS (row 5) for all gases is close to 1 which is primarily due to the T–Ph approach and the selection of the regularization parameter $\alpha$ based on minimizing the intraday variability of the gas TCs. Row 6 of Table 3 shows the value of the trend estimate. To assess the trend values, the methodology described by (Gardiner et al., 2008) was used (see below) which is based on the RMS approximation of the gas concentration variability by a three–term segment of the Fourier series and bootstrap methods of confidential intervals assessment for 95 % probability. Finally, row 7 shows the RMS difference between the Xgas and the trigonometric Fourier series used to estimate their temporal variability. For CFC-11 and CFC-12, these values are close to the estimate of the random error ($2.8$ vs $3.08\,\%$ and $2.1$ vs $2.40\,\%$) which indicates an adequate description of their variability by the Fourier series. At the same time, for HCFC-22, the RMS difference is $5.3\,\%$ which exceeds the random error of $3.7\,\%$, and the HCFC-22 variability contains some other components besides the trigonometric Fourier series. The reason for such behavior of HCFC-22 is a reduction of its use during the period analyzed that leads to a decrease in its growth rate. As a result, the representation of its variability in a form of a linear increase and seasonal variations, represented by trigonometric Fourier series (see Section 3.2), cannot be accurate. Polyakov et al. (2020b) analyzed the decrease in a growth rate of HCFC-22 abundancies over St. Petersburg in the past decade.

**Table 3.** Summary of the statistics for retrieved halocarbons TCs after filtering

| N | Parameter | CFC-11 | CFC-12 | HCFC-22 |
|---|-----------|--------|--------|---------|
| 1 | Number of spectra/days | 3864/678 | 3912/664 | 3855/663 |
| 2 | $RMS(\chi^2)$ | $0.52 \pm 0.18$ | $0.40 \pm 0.16$ | $0.34 \pm 0.13$ |
| 3 | Mean TC, $cm^{-2}$(Xgase, pptv) | $4.75 \times 10^{15}(225)$ | $10.42 \times 10^{15}(493)$ | $5.04 \times 10^{15}(238)$ |
| 4 | intraday SD of Xgas, % | 0.76 | 0.58 | $3.74(4.54/2.32)^*$ |
| 5 | DFS | $1.05 \pm 0.06$ | $1.20 \pm 0.05$ | $1.00 \pm 0.00$ |
| 6 | Trend, $\% \, yr^{-1}$ | $-0.40 \pm 0.07$ | $-0.49 \pm 0.05$ | $2.12 \pm 0.13$ |
| 7 | Total SD of Xgas, % (except Fourier app.) | 2.8 | 2.1 | 5.3 |

$^*$before / after February 2016

The SFIT4 software allows to calculate an error budget based on the Rodgers (2000, Chapt. 3) approach for each measurement. Rodgers (2000, Eq. 3.16) considers 4 components of the measurement error: the smoothing error, model parameter error, forward model error, and the retrieval noise. To estimate the mean smoothing error, it is necessary to have real covariance matrices of the gas vertical profiles, which are not available, therefore we cannot estimate this component of the error. We can only assume that it is small, because due to their long lifetime, we expect constant VMR profiles of the target gases in the troposphere. Prignon et al. (2019) showed that the smoothing error for HCFC-22 is rather small (0.3 %). The model parameter error is caused by the inaccuracies in setting the parameters describing the instrument and the state of the atmosphere.

To calculate the terms of the model parameter error, which are shown in rows 1–7 of Table 4, equation (Rodgers, 2000, Eq. 3.18) was used. For this equatuon, it is necessary first to set the uncertainties of various parameters which are taken into account. Rodgers (2000) enters them as elements of the $S_b$ matrix, the corresponding column for these elements is presented in Table 4. For the temperature profile (row 1) below $40 \, km$, where the profiles of the target gases are derived, the absolute value of the temperature systematic error totals $1 - 2 \, K$, random error totals $2 - 4 \, K$ depending on altitude. For other parameters, the relative errors are indicated in other rows of the Table. In addition to the fixed parameters, two types of parameters were derived in the retrieval process: 1) the retrieval parameters including a number of instrumental parameters such as BSS (a slope for all three gases and a curvature for CFC-11), along with instrumental line shape, channeling before 2016, zero level uncertainty, etc., and 2) the content of interfering atmospheric gases listed in Table 1 (column "other gases"), which absorption lines overlap with lines of the target gas. Their contribution to the errors of target gases TCs retrieval are shown in rows 8 (Interfering species) and 9 (Retrieval parameters). The forward model error (Rodgers, 2000, Eq. 3.16) was considered to be negligible in our retrieval. The retrieval noise shown in row 10 indicates the error related to the spectra measurement noise.

A row 11 in Table 4 demonstrates that the total systematic errors of the TCs retrieval for CFC-11 and HCFC-22 are relatively large, amounting to 7.61 and 5.75 %, and these values are almost entirely due to the uncertainty in the spectroscopic information on the intensities of the pseudo–lines (row 3 of Table 4). For CFC-12, the total systematic error is estimated at 2.2 %, the main source of this error is the uncertainty of the temperature profile (row 1 of Table 4). Note that the value of the total systematic

error is slightly variable, its SD is maximal for CFC-11 comprising for $0.16\%$. The total random error as well as its variability is maximal for HCFC-22. The main contribution to it is made by the spectral measurement error which is caused by the low

365 absorption of the solar radiation by this gas. The random components of the total error for other two gases, $3.08$ and $2.40\%$, are more stable and their main source is the error in the temperature profile (see row 1). It should be noted that the filter used has a significant effect on the errors and variability of the HCFC-22 TCs. When switching to the IRWG NDACC f6 filter in February 2016, the intraday variability of the results (row 4 of Table 3) has decreased by approximately 2 times, and the random component of the total error (row 11 of Table 4) has decreased by $1.4\%$, mainly due to the error of spectroscopic

370 measurements. Due to channeling, the F3 filter (used before February 2016), leads to a large scatter in retrieval results owing to larger errors (see section 2.1).

**Table 4.** Error budget for retrieved halocarbons TCs. The relative uncertainties of spectroscopic parameters are $7, 1$ and $5\%$ for CFC-11, CFC-12, and HCFC-22, respectively. $S_b$ means a priori imprecision of parameters.

| N | Gas | | CFC-11 | | CFC-12 | | HCFC-22 | |
|---|---|---|---|---|---|---|---|---|
| | | | | | TCs error, % | | | |
| | Parameter | $S_b$, % | Systematic | Random | Systematic | Random | Systematic | Random |
| 1 | Temperature | | $2.29 \pm 0.25$ | $2.56 \pm 0.30$ | $1.96 \pm 0.15$ | $1.96 \pm 0.12$ | $1.72 \pm 0.07$ | $1.50 \pm 0.06$ |
| 2 | SZA | $0.1 \pm 0.5$ | $0.20 \pm 0.17$ | $1.03 \pm 0.84$ | $0.22 \pm 0.18$ | $1.09 \pm 0.89$ | $0.25 \pm 0.25$ | $1.27 \pm 1.27$ |
| 3 | Target line intensity | 7/1/5 | $7.02 \pm 0.28$ | | $0.45 \pm 0.49$ | | $5.04 \pm 0.45$ | |
| 4 | Target temperature dependence of line width | 7/1/5 | $0.00 \pm 0.00$ | | | | $0.27 \pm 0.05$ | |
| 5 | Target air broadening of line width | 7/1/5 | $0.02 \pm 0.03$ | | $0.61 \pm 0.14$ | | $2.16 \pm 0.24$ | |
| 6 | H$_2$O spectroscopy | 10 | $1.45 \pm 0.57$ | | $0.31 \pm 0.31$ | | $0.25 \pm 0.35$ | |
| 7 | zshift | $1 \pm 1$ | $1.03 \pm 0.10$ | $1.03 \pm 0.10$ | $0.12 \pm 0.01$ | $0.25 \pm 0.03$ | $0.10 \pm 0.01$ | $0.20 \pm 0.02$ |
| 8 | Interfering species | | $0.04 \pm 0.04$ | | $0.02 \pm 0.01$ | | $0.19 \pm 0.12$ | |
| 9 | Retrieval parameters | | $0.12 \pm 0.07$ | | $0.02 \pm 0.00$ | | $0.29 \pm 0.05$ | |
| 10 | Spectra measurement noise | | | $0.29 \pm 0.13$ | | $0.20 \pm 0.03$ | | $2.66 \pm 1.62$ $(3.3/1.8)^*$ |
| 11 | Total | | $7.61 \pm 0.16$ | $3.08 \pm 0.36$ | $2.24 \pm 0.14$ | $2.40 \pm 0.54$ | $5.75 \pm 0.08$ | $3.70 \pm 1.29$ $(4.32/2.92)^*$ |

$^*$before / after February 2016

## 3.2 Analysis

Figures 5–7 show the results obtained in a form of the daily means of both the TCs and Xgas. The TC values directly represent the results of spectra inversion, while the Xgas are calculated by dividing the gas total column by the dry air total column. The analysis of the retrievals on a form of Xgas values prevents the influence of the surface pressure and humidity variability and thus are more stable. To analyze the variability of the target gases on the scale of both long–term trends and seasonal variability, we used the approach implemented by Gardiner et al. (2008) for assessing the trends which is based on the approximation of a series of data by expansion in a finite–dimensional basis, Eq. (1).

$$F(t) \approx a + bt + c_1 f_1(t) + c_2 f_2(t) \ldots + c_k f_k(t) \tag{1}$$

In Eq. (1), $F(t)$ is the dependence approximated by the expansion, in our case represented by discrete measurement data, $t$ – time (years), $a$ – constant, $b$ – linear term coefficient that is equal to trend value, $c_i$, $\quad i = 1, k$ – coefficients, $k$ – number of the coefficients, $f_i(t)$ – basis functions. Due to the annual cyclical nature of atmospheric processes, a trigonometric Fourier series with a maximum period of a year is used which correspond to the basis functions (2)

$$f_{2i-1}(t) = cos(2\pi it), \qquad f_{2i}(t) = sin(2\pi it), \qquad i = 1, m \tag{2}$$

for m = 3 or, which is the same, k = 6. Let us write Eq. (1) in the form (3), highlighting the nonlinear part S(t) (4) .

$$F(t) \approx a + bt + S(t) \tag{3}$$

$$S(t) = c_1 f_1(t) + c_2 f_2(t) \ldots + c_k f_k(t) \tag{4}$$

$S(t)$ can be considered as a periodic component of the measurement data time sequence and its one period can be analyzed as a seasonal data variability. Figures 5–7 in addition to the daily mean values of Xgas and TC also show a dashed straight linear trend $a + bt$ and, with a solid black line, the result of approximating the measurement data by the trigonometric Fourier series (1), (2). Fig. 5 demonstrates a pronounced periodicity of the results, showing the seasonal variation of both TCs and Xgas of CFC-11. As it is shown below, a similar periodicity is also observed in satellite measurements and in the WACCM data. As expected, the Xgas exhibit a slightly smaller scatter than the TCs. Note that starting from April 2019, there is a sharp increase in the concentrations of CFC-11. At present, we have no way to explain whether such growth is objectively presented or caused by peculiarities in the operation of the device. At the same time, this growth noticeably affects the trend estimates. Therefore, when calculating the trends of the CFC-11 TCs and Xgas, we limited CFC-11 time series to April 2019, leaving the analysis of the reasons for this feature outside the scope of this study.

Analysis of the CFC-12 measurements (Fig.6) shows significantly different results. First of all, the comparison of Fig. 5 and 6 and estimates of the intraday variability and measurement uncertainties of these gases demonstrate that the TCs and Xgas of CFC-12 show less scatter than that of CFC-11 except for some isolated anomalies. The seasonal variability of these values for

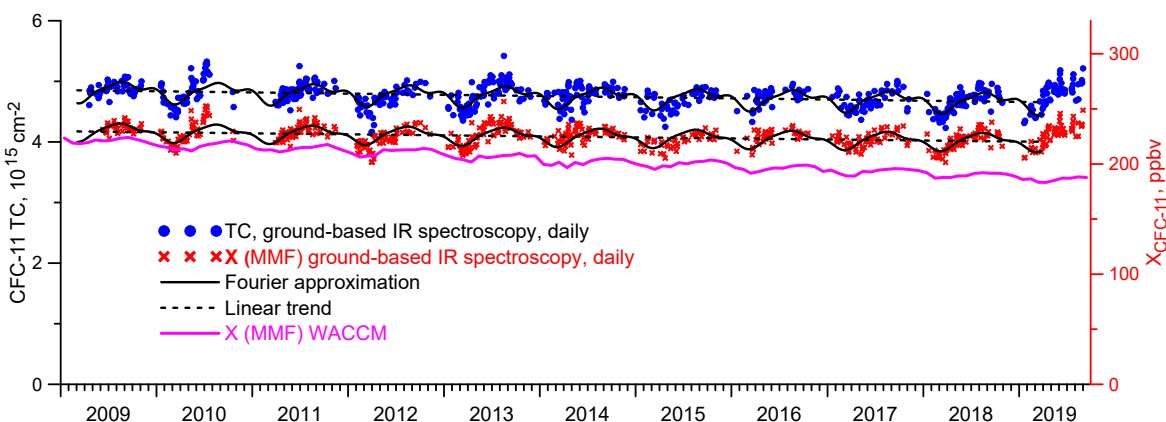

**Figure 5.** Daily mean TCs and Xgas of CFC-11. T–Ph parameter $\alpha = 85$.

CFC-12 is noticeably less than that of CFC-11. We also note that moving from TCs to Xgas, the deviations of the results from the approximating segment of the Fourier series decrease significantly. That is, variations in surface pressure and water vapor TCs make a significant contribution to the variability of the CFC-12 TCs which indicates small changes in its VMR profile. We will analyze these factors in detail in the next Subsection 3.3.

Having considered the results of measurements of HCFC-22 daily means, we observed a large variability consistent with a large random component of the total error estimates (see Table 4). There are also noticeable seasonal variations. At first glance, the filter change in February 2016 clearly manifests itself in a change in the data scatter, but in 2016 the scatter looks no less than in previous years, sharply decreasing in 2017 and later. Noteworthy is the observed cessation of the increase in HCFC-22 values starting from 2018 previously described by Polyakov et al. (2020b). We also observed an increase in the scatter of the results for all three gases in 2013 due to a decrease of the SNR values caused by degradation of the tracking system mirror.

Table 5 presents trend estimates for Xgas time series using two different methods described by Gardiner et al. (2008) and Timofeev et al. (2020). Gardiner et al. (2008) model the intra-annual variability in terms of a Fourier series, Timofeev et al. (2020) use monthly mean values of the considered period to describe a seasonal cycle. In both methods, trends were estimated

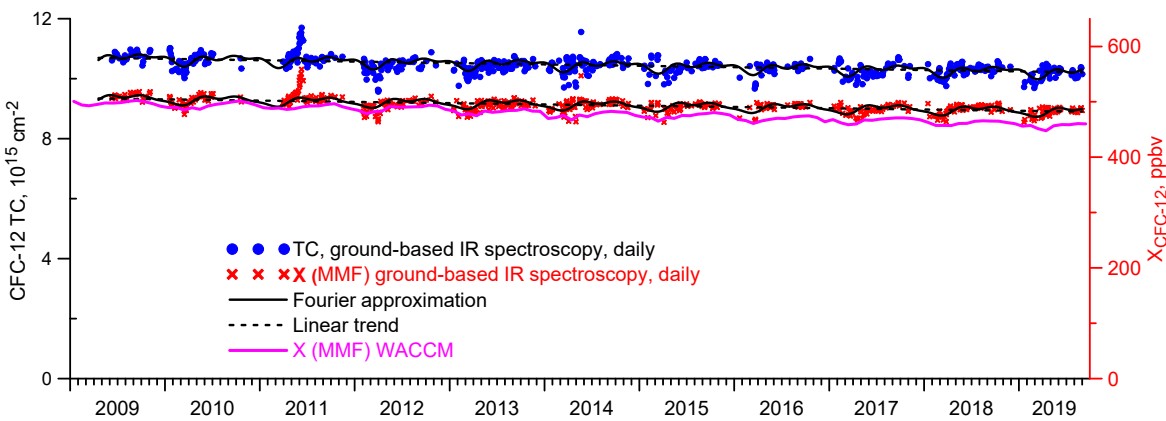

**Figure 6.** Daily mean TCs and Xgas of CFC-12. Filtered results, T-Ph parameter $\alpha$=85.

410    by subtracting the seasonal variability from initial time series. In first method, we considered periodicities of 4 month and larger, in second method, monthly mean values accounted for periodicities from 1 month. The estimation of the width of the confidence interval of the trend value for the Gardiner's approach was carried out using the Bootstrap method, for the Timofeev's method, it was calculated on the basis of a theoretical statistical approach. It is worth mentioning that we did not take into account the autocorrelation that could be presented in long-lived Xgas time series. Santer et al. (2000) demonstrated

415    that neglecting of the autocorrelation in time series can affect the trends estimates and underestimate uncertainties, however due to substantially irregular FTIR measurements it was difficult to estimate it.

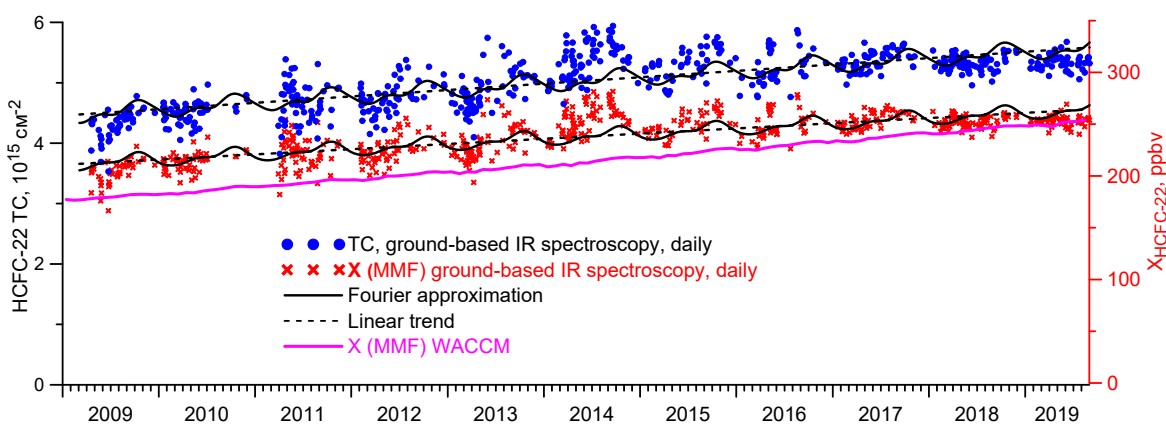

**Figure 7.** Daily mean TCs and Xgas of HCFC-22. Filtered results, T-Ph parameter $\alpha = 3 \times 10^3$.

**Table 5.** Estimated trends of the halocarbon Xgas derived from the FTIR measurements at the St. Petersburg site in 2009–2019.

| Gas | Gardiner et al (2008) | Timofeyev et al (2020) |
|---|---|---|
| CFC-11 | -0.40±0.07 | -0.39±0.08 |
| CFC-12 | -0.49±0.05 | -0.46±0.05 |
| HCFC-22 | 2.12±0.13 | 2.22±0.14 |

Table 5 demonstrates that the differences between two methods remain within the 95 % confidence intervals, i.e. are not significant.

### 3.3 Comparison with independent data

We compared the FTIR results at the St. Petersburg site with the data of measurements and modeling. There were no available experiments that are fully consistent with our FTIR measurements in space and time, so we tried to find the closest possible data. There are three sources of data on the concentration of the halocarbons in the atmosphere. First source – the in situ measurements at the surface (carried out by exactly in situ and the flask methods) are available from the AGAGE (Dunse et al., 2005) and HATS (Montzka et al., 1993) observational networks, data are regularly updated at ftp://ftp.cmdl.noaa.gov/hats/hcfcs/hcfc22/flasks/. Measurements are carried out at the fixed locations, the closest of which is Mace Head, Ireland (MHD) at a distance of $2500\,\mathrm{km}$ and $6.6°$ south of the St. Petersburg site. The mean values and the trends of the Xgas are calculated from the FTIR data and from the MHD site near–ground data for the period (2009–2019) for all three gases. The results are shown in Table 6 in columns 1 and 3. The mean near–ground VMRs (GVMRs) at the MHD site for CFC-11, CFC-12, and HCFC-22 equal 234, 517, and 237 pptv, whereas the FTIR Xgas measurements show 225, 493, and 252 pptv means, respectively. The trend values (Table 6, columns 2, 4) for the GVMR data are $-0.53$, $-0.59$, $2.0\,\%\,\mathrm{yr}^{-1}$, for the FTIR data are $-0.38$, $-0.48$, $2.0\,\%\,\mathrm{yr}^{-1}$ for CFC-11, CFC-12, and HCFC-22, respectively. Taking into account the spatial discrepancy, the different nature of the measured quantities, and different measurement conditions (background conditions on the Atlantic coast and measurements near the large agglomeration of St. Petersburg), the agreement between the mean values and the trends can be considered satisfactory. It should be noted that the differences in trend estimates do not go beyond the differences in trend values obtained by other researchers. Zhou et al. (2016) have obtained trends of $-0.86$, $-0.76$, $2.84\,\%\,\mathrm{yr}^{-1}$ for the period 2009–2016; the WMO (2018) indicated that averaged VMRs for 2015 comprised for $229.2-231.1$, $515.3-519.7$, $233.0-238.0$ pptv and the trends for the period 2010–2016 were $-0.70$, $-0.47$, $2.54\,\%\,\mathrm{yr}^{-1}$ for CFC-11, CFC-12, and HCFC-22, respectively. Taking into account the decrease in both the rate of decay of CFC-11 and the rate of growth of HCFC-22 (e.g. Polyakov et al., 2020b), the agreement of both concentrations and trend values seems to be satisfactory.

The second source of information on the halocarbons content is the satellite measurements, most fully for the target gases presented by the ACE–FTS instrument data which version 4 is described by Boone et al. (2020). The ACE–FTS is a high spectral resolution $(0.02\,\mathrm{cm}^{-1})$ Fourier transform spectrometer operating from 2.2 to $13.3\,\mathrm{\mu m}$ $(750-4400\,\mathrm{cm}^{-1})$ based on a Michelson interferometer. The instrument is a main payload onboard SCISAT–1 satellite with drifting orbit, inclination $73.9°$, and altitude $750\,\mathrm{km}$. Working primarily in solar occultation mode, the satellite provides vertical profile information (typically $10-100\,\mathrm{km}$) for temperature, pressure, and the VMRs of dozens of atmospheric gases over latitudes $85°\,\mathrm{N}$ to $85°\,\mathrm{S}$. The lower boundary of the retrieved profiles does not fall below $6\,\mathrm{km}$, but, as a rule, is above $7-8\,\mathrm{km}$, and the errors at the lower level may be greater than in the rest of the profile. Therefore, we used for comparisons only the profiles in which the data were available above $7\,\mathrm{km}$ and analyzed the average satellite VMRs (SVMR) in the $8-12\,\mathrm{km}$ layer to reduce the random error. For comparison, we selected the ACE–FTS measurements closer than $500\,\mathrm{km}$ from the St. Petersburg site. In 2009–2019, there are only 47 days of the SVMR measurements for CFC-11 (mean 233 pptv, trend $-0.68\pm0.23\%\,\mathrm{yr}^{-1}$), 47 days for CFC-12 (mean 521 ppyv, trend $-0.52\pm0.16\,\%\,\mathrm{yr}^{-1}$), and 46 days for HCFC-22 (mean 240 pptv, trend $2.0\pm0.5\,\%\,\mathrm{yr}^{-1}$). The results are shown in columns 7 and 8 of Table 6. Due to the peculiarity of the orbit and the weather conditions at the St. Petersurg

site SCISAT–1 measurements are available on rare occasions; during 10 years we have found not more than 47 measurements closer than $500\,\mathrm{km}$ from the St. Petersburg site. However, the $95\,\%$ probability intervals show reliability of the trend estimates using the bootstrap method by Gardiner et al. (2008). As one can see by comparing columns 1 and 7 of Table 6, the confidence intervals of the means overlap, i.e. the difference in the mean values is not significant only for HCFC-22, and for both CFC-11 and CFC-12, the SVMR is significantly greater than the FTIR Xgas, the difference totals $8\,\mathrm{pptv}$, or $3.5\,\%$ for CFC-11 and $28\,\mathrm{pptv}$ or $6.3\,\%$ for CFC-12. To increase the number of data pairs, we analyzed all ACE–FTS data at all longitudes in the $55-65°\,\mathrm{N}$ latitudinal range including the St. Petersburg site (about $60°\,\mathrm{N}$). For the period of the FTIR measurements, the SVMR data contains 1113 measurements for CFC-11, with a mean of $235.3\,\mathrm{pptv}$ and a trend value of $-0.63\,\%\,\mathrm{yr}^{-1}$, 1120 measurements for CFC-12 ($526.4\,\mathrm{pptv}, -0.58\,\%\,\mathrm{yr}^{-1}$), and 1111 measurements for HCFC-22 ($239.5\,\mathrm{pptv}, 2.2\,\%\,\mathrm{yr}^{-1}$), see columns 5 and 6 of Table 6.

**Table 6.** The means and the trends estimates of the FTIR Xgase, GVMR and SVMR measurements, and the WACCM Xgas. If the width of the confidence interval is not specified, it is less than the last significant digit.

| Gas | IR spectroscopy, St. Petersburg, Xgase | | Mace Head, Ireland, GVMR | | SCISAT, mean VMR 8˘12 km, SVMR | | | | WACCM, WXgase St. Petersburg | |
| | Mean, pptv | Trend, $\%\,\mathrm{yr}^{-1}$ | Mean, pptv | Trend, $\%\,\mathrm{yr}^{-1}$ | Mean, pptv | Trend, $\%\,\mathrm{yr}^{-1}$ | Mean, pptv | Trend, $\%\,\mathrm{yr}^{-1}$ | Mean, pptv | Trend $\%\,\mathrm{yr}^{-1}$ |
| | | | | | 55–65°N | | Distance $<$ 500km | | | |
| | 1 | 2 | 3 | 4 | 5 | 6 | 7 | 8 | 9 | 10 |
| CFC-11 | 225 | $-0.40\pm0.07$ | 234 | $-0.53\pm0.02$ | 235 | $-0.63$ | $233\pm3$ | $-0.68\pm0.23$ | $203\pm2$ | $-1.68\pm0.06$ |
| CFC-12 | $493\pm1$ | $-0.49\pm0.04$ | 517 | $-0.59\pm0.01$ | 526 | $-0.58$ | $521\pm4$ | $-0.52\pm0.16$ | $478\pm2$ | $-0.84\pm0.03$ |
| HCFC-22 | 238 | $2.12\pm0.13$ | 237 | $2.0\pm0.05$ | 240 | $2.2$ | $240\pm7$ | $2.0\pm0.5$ | $215\pm4$ | $3.40\pm0.03$ |

With a 20 times larger dataset, the confidence intervals for the trends for the latitudinal belt are much narrower than for a circle with a radius of $500\,\mathrm{km}$, thus the differences in the SVMR trends vs FTIR Xgas trends for CFC-11 and CFC-12 become significant. Such discrepancy may be due to the different physical nature of the compared quantities. The satellite data does not take into account the lower tropospheric layers where the influence of anthropogenic pollution sources is the greatest. Therefore, analyzing the trends and proceeding from this that the background values of the VMRs for atmospheric CFC-11 and CFC-12 (in situ and satellite) are falling faster than the FTIR Xgas in the industrially developed European part of Russia (near megacity St. Petersburg), we may assume that some sources of CFC-11 and CFC-12 exist somewhere there. The absolute values of CFC-11 and CFC-12 FTIR Xgas are smaller than that of the in situ and satellite measurements, but this may only be due to the uncertainty of the used spectroscopy, see the estimates of the systematic in Table 4, row 3.

Figure 8 depicts the seasonal variation functions S (t), Eq. (4), for three gases and for four types of data: near–ground VMR at the MHD station (GVMR), satellite mean VMR $8-12\,\mathrm{km}$, $55-65°\,\mathrm{N}$ (SVMR), the Xgas by FTIR measurements (Xgas), and the Xgas from the WACCM (WXgase).

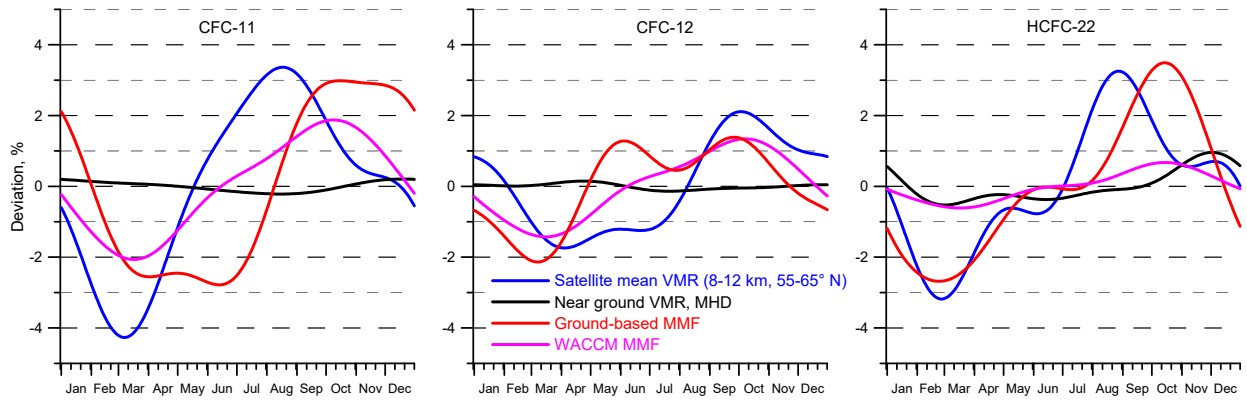

**Figure 8.** Seasonal relative variability of CFC-11 (left), CFC-12 (center), and HCFC-22 (right). Satellite refer to the ACE-FTS measurements, ground-based and WACCM refer to the FTIR measurements and numerical modeling at the St. Petersburg site.

There are some fundamental differences between local surface and remote sensing measurements (satellite and ground-based FTIR). First, surface measurements are performed with a high regularity and frequency, resulting in stable averages. Secondly, they are unaffected by variations in pressure and tropopause height. And finally, the surface data used were obtained in close to background conditions. Therefore, Figure 8 demonstrates the low seasonal variability of the GVMR – within tenths of a per cent for CFC-11 and CFC-12 and within $0.7\%$ for HCFC-22. At the same time, a noticeable seasonal variation of the FTIR Xgas and the SVMR values for all three gases and of the WACCM Xgas for CFC-11 and CFC-12 are observed. The maximum amplitude of the variability for CFC-11 reaches $4\%$, for HCFC-22 slightly exceeds $3\%$, and for CFC-12 is close to $2\%$. For all three gases, the seasonal variations of the SVMR and the Xgas are qualitatively and quantitatively similar: in spring (March–April) there is a minimum, and in late summer or autumn (August–October) is a maximum. At the same time, there are some differences in the seasonal cycles: for CFC-11, the change in the Xgas is 2-3 months ahead of the SVMR,

while for HCFC-22 the autumn maximum shows the same tendency, whereas the spring minimum, on the contrary, is observed simultaneously for the Xgas and the SVMR. For CFC-12, the Xgas amplitude is approximately half that for the two other gases; the spring minimum of the Xgas, on the contrary, is observed before that of the SVMR, and the autumn maxima are coincided. For CFC-12, a second maximum in the Xgas seasonal cycle is observed in the early summer. The WXgas for CFC-11 and CFC-12 in a whole show a qualitatively and quantitatively similar character of seasonal variability to the Xgas and the SVMR, while for HCFC-22, on the contrary, the changes in the WXgas are significantly less (less than $1\%$) than that of the Xgas and the SVMR. In general, we can conclude that the Xgas, SVMR, and WXgas show qualitative similar seasonal variation with some quantitative differences, and the GVMR variabilities are significantly less. The variability of the WXgas for HCFC-22 only depicts the exception, it is essentially less than the Xgas and SVMR variability.

## 4 Conclusions

1. The retrieval strategies for deriving the TCs of CFC-11, CFC-12, and HCFC-22 using ground-based IR solar spectra measurements by Bruker IFS125HR spectrometer at the St. Petersburg site were improved. For solving the inverse problem, values of regularization parameter of the T–Ph approach were optimized. For the FTIR measurements over the NDACC site St. Petersburg in 2009–2019, the estimates of the DFS values are $1.05 \pm 0.06$, $1.20 \pm 0.05$, $1.00 \pm 0.00$, the estimates of the relative systematic and random errors are $7.61\%$ and $3.08\%$, $2.24\%$ and $2.40\%$, and $5.75\%$ and $3.70\%$, for CFC-11, CFC-12, and HCFC-22, respectively.

2. The time series of the TCs and Xgas for CFC-11, CFC-12, and HCFC-22 above the St. Petersburg site near Saint-Petersburg, Russia in 2009–2019 were obtained. Mean values of $X_{CFC-11}$, $X_{CFC-12}$, and $X_{HCFC-22}$ are 225, 493, and 238 pptv. The RMS intraday variability of the TCs of measured gases are 0.8, 0.6, 2.3 $\%$ for three gases in the same order. Estimates of the Xgas trends of CFC-11, CFC-12, and HCFC-22 equal $-0.40 \pm 0.07\ \%\ yr^{-1}$, $-0.49 \pm 0.05\ \%\ yr^{-1}$, and $2.12 \pm 0.13\ \%\ yr^{-1}$, respectively. The analysis of the seasonal variability of CFC-11, CFC-12, and HCFC-22 demonstrated the similar qualititaive seasonal variability for all three gases with minimum in spring–begin of summer and maximum in fall; $X_{CFC-11}$ and $X_{HCFC-22}$ variability amounts to $3\%$, the variability of $X_{CFC-12}$ amounts to $2\%$.

3. Mean values, trends and seasonal variability of $X_{CFC-11}$, $X_{CFC-12}$, and $X_{HCFC-22}$ above St. Petersburg site were compared to the same parameters of the near ground VMRs measured at the site Mace Head, Ireland. It is shown that the mean of the $X_{CFC-11}$ above St. Petersburg site is 9 pptv$(3.8\%)$ less than the mean GVMR at MSH site, the mean of the $X_{CFC-12}$ is 24 pptv$(4.6\%)$ less than the mean GVMR, and the mean of the $X_{HCFC-22}$ does not significantly differ from the mean GVMR. In ablolute values, the trend of the $X_{CFC-11}$ is 0.13 $\%\ yr^{-1}$ less ($-0.40$ vs $-0.53\ \%\ yr^{-1}$), the trend of the $X_{CFC-12}$ is 0.10 $\%\ yr^{-1}$ less ($-0.49$ vs $-0.59\ \%\ yr^{-1}$) than that of the GVMR, and the trend of the $X_{HCFC-22}$ does not significantly differs from that of the GVMR ($2.12 \pm 0.13\ \%\ yr^{-1}$ vs $2.0 \pm 0.05\ \%\ yr^{-1}$). The seasonal variability of the GVMR for all three gases is much lower than the Xgas variability.

4. Mean values, trends and seasonal variability of $X_{CFC-11}$, $X_{CFC-12}$, and $X_{HCFC-22}$ above St. Petersburg site were compared to the same parameters of the SVMR. The SVMR stands for the mean values of VMR, measured with ACE–FTS

between altitudes $8 - 12\,\mathrm{km}$ and between latitudes $55 - 65°\,\mathrm{N}$. It is shown that the mean $\mathrm{X_{CFC-11}}$ is $10\,\mathrm{pptv}(4.3\,\%)$ less than the mean SVMR, the mean $\mathrm{X_{CFC-12}}$ is $33\,\mathrm{pptv}(6.3\,\%)$ less than the mean SVMR, and the mean $\mathrm{X_{HCFC-22}}$ is $2\,\mathrm{pptv}(0.8\,\%)$ less than the mean SVMR. In absolute value, the $\mathrm{X_{CFC-11}}$ trend is $0.23\,\%\,\mathrm{yr^{-1}}$ less $(-0.40\,\mathrm{vs}-0.63\,\%\,\mathrm{yr^{-1}})$ than the trend of the SVMR, the trend of $\mathrm{X_{CFC-12}}$ is $0.09\,\%\,\mathrm{yr^{-1}}$ less than the trend of the SVMR $(-0.49\,\mathrm{vs}-0.58\,\%\,\mathrm{yr^{-1}})$, and the trend of $\mathrm{X_{HCFC-22}}$ does not significantly differ from that of the SVMR $(2.12\pm0.13\,\mathrm{vs}\,2.2\,\%\,\mathrm{yr^{-1}})$. The Xgas and SVMR show qualitative and quantitative similar seasonal variation.

5. Mean values, trends and seasonal variability of $\mathrm{X_{CFC-11}}$, $\mathrm{X_{CFC-12}}$, and $\mathrm{X_{HCFC-22}}$ were compared to the same parameters of the WXgas. The WXgas stands for the Xgas calculated on the basis of the WACCM dataset of the VMR profiles for the St. Petersburg site. It is shown that the mean $\mathrm{X_{CFC-11}}$ is $22\,\mathrm{pptv}\,(10\,\%)$ greater than the mean $\mathrm{WX_{CFC-11}}$ , the mean $\mathrm{X_{CFC-12}}$ is $15\,\mathrm{pptv}\,(3.1\,\%)$ greater than the mean $\mathrm{WX_{CFC-12}}$ , and the mean $\mathrm{X_{HCFC-22}}$ is $23\,\mathrm{pptv}\,(10\,\%)$ greater than the mean $\mathrm{WX_{HCFC-22}}$. In absolute value, the $\mathrm{X_{CFC-11}}$ trend is $1.28\,\%\,\mathrm{yr^{-1}}$ less than the trend of the $\mathrm{WX_{CFC-11}}$ $(-0.40\,\mathrm{vs}$ $-1.68\,\%\,\mathrm{yr^{-1}})$, the trend of the $\mathrm{X_{CFC-12}}$ is $0.35\,\%\,\mathrm{yr^{-1}}$ less than the trend of the $\mathrm{WX_{CFC-12}}$ $(-0.49\,\mathrm{vs}-0.84\,\%\,\mathrm{yr^{-1}})$, and the trend of the $\mathrm{X_{HCFC-22}}$ is $1.28\,\%\,\mathrm{yr^{-1}}$ less than the trend of the $\mathrm{WX_{CFC-22}}$ $(2.12\,\%\,\mathrm{yr^{-1}}\,\mathrm{vs}\,3.40\,\%\,\mathrm{yr^{-1}})$. The Xgas and WXgas show qualitative and quantitative similar seasonal variations for CFC-11 and CFC-12; the seasonal variability of the $\mathrm{WX_{HCFC-22}}$ is essentially less than the $\mathrm{X_{HCFC-22}}$ variability.

In general, the comparison of the FTIR Xgas with the independent data shows a good agreement of their means within the systematic error of the measurements. The trends observed over the St. Petersburg site demonstrate the smaller decrease rates for CFC-11 and CFC-12 than the independent data, and the same increase rate for HCFC-22. The retrieval techniques suggested may be used in further development of unified strategies for halocarbons retrieval at the sites of the NDACC observational network that use the Bruker IFS125HR spectrometers for IR solar spectra measurements.

*Data availability.* The FTIR CFC-11, CFC-12 and HCFC-22 retrievals at St Petersburg are available from NDACC (http://www.ndaccdemo.org/data). The MHD HATS CFC-11, CFC-12, and HCFC-22 data are publicly available from NOAA (http://www.esrl.noaa.gov/gmd/hats. SCISAT-1 ACE-FTS data version 4.1 can be asked from ACE-FTS group (http://www.ace.uwaterloo.ca/data.php). WACCM profiles at St. Petersburg site are available from IRWG NDACC (https://www2.acom.ucar.edu/irwg/links).

## Appendix A: Water vapor continuum

Figure A1 depicts the transmission functions of water vapor continuum. Although sometimes a value of precipitable water (PW) above St.Patersburg reaches $50\,\mathrm{mm}$, in clear-sky days, when FTIR measurements are performed, maximum value of PW is about $30 - 40\,\mathrm{mm}$.

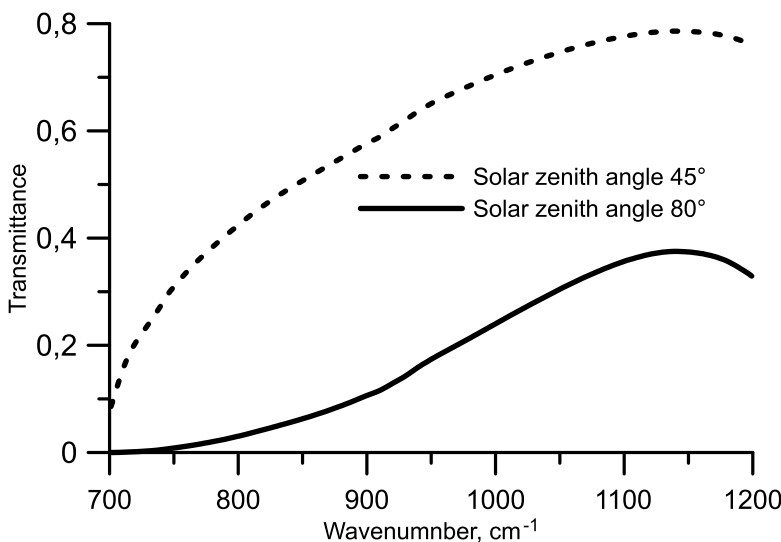

**Figure A1.** Transmission function of water vapor continuum for July 29, 2018 - one of the humid days of measurements in 2018 (PW totals 30 mm).

**Appendix B:  Results filtering**

On the next step we filtered the retrieved TCs using the following criteria: SNR of measured spectra should be in the range of 50 - 600, and the deviation from the mean statistical characteristics presented in Table 2 should not be greater than 2×SD. The used criteria and the percentage of the measurements discarded after their application are shown in Table B1.

**Table B1.** The criteria used and the percentage of data discarded after their application.

| Criterion | CFC-11 | | CFC-12 | | HCFC-22 | |
|---|---|---|---|---|---|---|
| | Value | Excluded, % | Value | Excluded, % | Value | Excluded, % |
| Sys err | 7.96 | 2.9 | 2.58 | 2.5 | 5.91 | 0 |
| Ran err | 4.77 | 5.0 | 4.42 | 4.7 | 9.50 | 4.8 |
| $\chi^2$ | 1.449 | 0.5 | 1.540 | 0.3 | 0.971 | 4.1 |
| DFS | 0.89 | 0 | 1.10 | 1.3 | 0.90 | 0 |
| S/N | $50-600$ | 7.9 | $50-600$ | 3.7 | $60-600$ | 4.2 |
| Not conv | Yes | 0 | Yes | 8.1 | Yes | 3.7 |
| div | Yes | 3.8 | Yes | 0 | Yes | 3.0 |
| No result | No files | 4.2 | No files | 1.2 | No files | 0 |
| Total excluded | 19 % | | 18 % | | 16 % | |
| Spectra/Days before filtering | 4773 / 720 | | 4768 / 718 | | 4585 / 714 | |
| after filtering | 3864 / 678 | | 3912 / 664 | | 3855 / 663 | |

Rows of Table B1:

1) systematic error (mean plus 2 SD)

2) random error (mean plus 2 SD)

3) residual (Xi2) (mean plus 2 SD)

4) DFS (mean minus 2 SD)

5) To exclude noisy spectra and possible non–linearity in measurements, we use only measurements with SNR values ranging from 50 (60) to 600.

6) Not converged

7) Divergence warning

8) SFIT did not present results

*Author contributions.* APol Conceptualization, Data curation, Formal analysis, Funding acquisition, Investigation, Methodology, Project administration, Resources, Software, Validation, Visualization, Writing – original draft. APob Investigation (performed measurements of spectral sensitivity functions, and was responsible for the measurements of solar radiation), Conceptualization (detected the effects of AWI). MM Investigation (was responsible for the measurements of solar radiation), Resources, Writing – review and editing. YV Investigation (the water vapor profiles), Writing – review and editing. YM Conceptualization (originally proposed a general research topic ).

*Competing interests.* The authors declare that they have no conflict of interest.

*Disclaimer.* TEXT

*Acknowledgements.* This work was supported by Russian Foundation for Basic Research, grant 18–05–00426. The ACE mission is funded by the Canadian Space Agency. NOAA data were provided by S. Montzka, 2019. We thank J. W. Hannigan (NCAR, Boulder, CO, USA) for providing the data of the WACCM model at the NDACC station in St. Petersburg. The ground–based TCs measurements of solar radiation at St. Petersburg site were obtained using the equipment of the SPbU "Geomodel" Resource Center.

570

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
