# Peer review of "Measurements of CFC-11, CFC-12, and HCFC-22 total columns in the atmosphere at the St. Petersburg site in 2009-2019"

_Atmospheric Measurement Techniques, 2020_

## Referee Comment (RC1) · Anonymous Referee #1 · 30 Nov 2020

Summary

This manuscript presents time series of three major long-lived halocarbons (CFC-11, CFC-12 and HCFC-22), as derived from ground-based Fourier Transform Infrared measurements performed at the high latitude low altitude site of Peterhof, in the vicinity of St Petersburg (SPB), Russia. A Tikhonov –Phillips regularization is adopted and the alpha parameter is selected for each gas such as to minimize intraday variability. A complete detailed uncertainty budget for the systematic and random components is carefully established and discussed.

The resulting decadal time series (2009-2019) are investigated in terms of absolute

atmospheric content, trend and seasonality. Comparisons with WACCM model data, satellite and surface in situ measurements are presented, and the authors conclude that a reasonable agreement is observed between all data sets.

This is a relevant contribution to AMT, and these time series will contribute to the verification of the success of the Montreal Protocol. This has been shown to be very important, even for species showing a decreasing trend, for which prohibited production has taken place recently (e.g., Montzka et al., 2018; Rigby et al., 2019).

However, some of the conclusions dealing with the regularization are not supported by the material currently presented, and as such, they could be seen as overstatements. Moreover, these conclusions are formulated as of general application. While the conclusions reached are almost certainly relevant for the SPB site, different situations might be encountered at other FTIR stations. Related concerns are detailed below.

I would recommend publication after careful revision of the corresponding sections, or removal of some of the discussions which are currently part of section 2.3 (and related sentences in the summary and conclusion sections).

Major concerns

As indicated above, the major concern with this contribution is that some of the conclusions are presented as universal truth while they are not backed by the material presented. These conclusions are all related to the choice of a Tikhonov-Phillips (T-Ph) regularization that would be more appropriate than OE, e.g.:

– T-Ph regularization it is more suitable for long-lived gases with a pronounced trend
– the profiles retrieved by the OE method are less reliable, showing distorted profile shapes when compared to T-Ph results

Although the T-Ph approach and its tuning are discussed in length in the manuscript, important information is missing regarding the OEM retrievals. No details are given

(or even summarized) on the choice of the many OEM parameters, on the resulting information content (DFS), on the error budgets. And what about the efforts deployed and the indicators used to validate the definition of the OEM a priori states for the three targets? A paper by the same authors listed in the references (Polyakov at al., 2018) might provide these details, but unfortunately this work is not freely available to me. If we miss this kind of information, we cannot be convinced that T-Ph is necessarily a better option than OEM. It might well be that non-optimum OEM results are compared with carefully tuned T-Ph products. Then it is potentially comparing apples and pears in Figures 2 to 4.

Regarding these figures, it is also important to keep in mind that total columns are retrieved, or a single piece of information (DFS is merely larger than 1). Then the shape of the retrieved profiles is not very relevant, especially because the computed error bars affecting each single (and meaningless taken alone) mixing ratio are so large that they define a broad range of possible solutions with an identical total column. A profile with a more realistic form could likely be drawn without exceeding the resulting uncertainty ranges.

Another reason given by the authors for selecting T-Ph against OEM regularization is the lack of supporting geophysical information needed to build a covariance matrix: "it should be taken into account that the OE approach requires the use of the covariance matrices for describing the variability of the target gases profiles, preferably the real covariance matrices, that are unavailable for the considered freons". However, 16 years of ACE profiles are available, they could be used (and consistently extrapolated down to SPB altitude) to construct it, including for the extra diagonal elements. MIPAS data products are also available to the community (see Chirkov et al., 2016). Then at least two multi-year data sets with global coverage. Second, OEM regularization could make use of ad hoc parameters such as to determine the a priori states. This might sound as artificial, but it is not that different than setting up the alpha for T-Ph. It is true however that it could be trickier, because several parameters are available: the per-layer a priori

covariance, the type of inter-layer correlation (Gaussian, exponential), the correlation length. Note by the way that for long-lived and well-mixed gases, it could be relevant to set up a length of several kilometers in order to avoid oscillations in the retrieved profiles.

Finally, it should be noted that the gases involved in the present study do not present "pronounced trends". CFC-11 and CFC-12 will have only changed by about 5% after 10 years. Even if it is larger for HCFC-22 ($\sim$25% over 2009-2019), this remains less than the changes observed for some other FTIR products over a single year. For example, ethane presents total columns varying on average by more than a factor two over a season, and still OEM regularization is fully and successfully applicable.

Addressing the concerns detailed above could in my opinion be done in two ways. First, provide some information on how the OEM regularization was optimized, and supply elements allowing the reader to evaluate and compare on more solid grounds the data products, in terms of uncertainty, daily scatter, DFS. . . Or to significantly reduce section 2.3 and to give focus on the valuable derived time series and the comparisons with other data sets. In any case, one should not be left with the message that T-Ph regularization is necessarily the good option for the retrievals of CFCs or HCFCs. Such conclusions cannot be reached with a single site study, especially when involving challenging conditions. It remains to be demonstrated that it would also be the case for other FTIR stations. So adding "for our site", "in our case" at some selected places in the text could be appropriate to temper the argumentation and make it more specific.

Second order issues and minor comments

I think the one-sentence description (starting line 42) of the impact of halogenated source gases on the formation of the ozone hole is a bit oversimplified and may deserve an additional statement informing about the roles of the stratospheric reservoirs of chlorine and of the heterogeneous reactions in chlorine activation. The gas phase

chemistry does not explain the massive polar ozone depletion, and the photolysis of CFCs does not happen in the Polar Regions since there is no high-energy UV photon available there. Another option is to identify and include a good reference.

Line 45: the Montreal Protocol does not act directly on the halocarbons emissions. Instead, the production of the relevant gases are limited and then banned. I suggest replacing "emission" by "production" on line 45.

Line 61: this sentence is somewhat misleading: the phase-out (100%) of the CFCs was decided in 1992, for a complete implementation by the end of 1995 (Copenhagen Amendment). In 1989, only a reduction was enforced by the initial treaty.

Line 62: Brown et al. 2011 is a good reference, but more recent trends have been published by the ACE team, considering now 16 years of measurement and improved versions of the data. I strongly suggest considering here and each time it is relevant these updated results (see Bernath et al., 2020), also for the trend comparisons.

Lines 63-66: it also depends on the evolution of the bromine and nitrogen stratospheric loadings!

Line 74: to my knowledge, the HFCs are targeted by the Kigali Agreement, not the HCFCs. HCFCs regulation is under the earlier amendments or adjustments. Please check and amend if needed.

Line 83: do you mean down to the Earth surface?

Line 86: it might be true that the publications on the subject were rather episodic, but not the measurements which were continuously performed and exploited at some of the NDACC sites. For example, halocarbon FTIR time series have been systematically included in the successive editions of the WMO assessments on ozone depletion.

Line 89: note that "freon" is a registered trademark.

Lines 93-94: this is not correct, Prignon et al. (2019) proposed an approach for the determination of total *and* partial columns, for the 1988-2017 time period (three decades), not only for 1999-2018.

Lines 106-108: I somewhat disagree with the arguments on the situation as to the absorptions and spectral signatures (quoted as low and smoothed). In my opinion, this discussion as presented misses the fact that the situation is very different for the three target gases: CFC-11 is likely the more difficult with a broad feature perturbed by strong water vapor lines; CFC-12 has a stronger and more isolated signature peaking at more than 10% and HCFC-22 presents a narrower feature quite free of interferences (HWHM probably on the order of ∼0.05 cm-1), resulting in the possibility to select a less wider micro-window. Of course, the spectral scenes will also be influenced by the latitude and altitude of the station.

Line 126: I guess QHN is another description of channeling? Note the relevant discussion paper on AMTD by Blumenstock et al. (2020). Harmonizing the designations could be helpful.

Line 163: the discussion on the spectral transmission function is interesting and original. If I understand correctly, the selection of the relevant parameters is conducted such as to limit the CFC-11 intraday variability. In the end, do you see any correlation between the water vapor and the CFC-11 total columns?

Lines 236-239: the statements regarding the Prignon et al (2019) paper are not correct. These authors also used a T-Ph regularization with alpha=9, minimizing the smoothing and measurement errors as per Steck (2002). Please amend your text accordingly.

[Figure]

Line 268: regarding the smoothing error, Prignon et al. (2019) have indeed evaluated it to be small (see Table 1 in their paper).

Line 297: is the mean molar fraction (or MMF) another name for the dry air mole fraction (often denoted xTARGET, see e.g., section 2.4 in Barthlott et al., 2015), as used by the NDACC and TCCON communities? If yes, it might be good here also to harmonize the designations.

Table 4 reports much higher daily SD for HCFC-22 than for other gases. Is this a filter 3 effect (before 02/2016)? Or is this just because intraday variability for an unregulated gas might be affected by polluted episodes (excursion above the baseline in in situ surface time series)? Also and with the exception of HCFC-22, the random errors quoted on line 18 are significantly larger than the intraday SD on line 4. Shouldn't they be commensurate?

Trends on Table 5: it is more and more clear that the uncertainties affecting trends are often underestimated because the methods used do not account for the auto-correlation present in the time series (see e.g., Santer et al., 2000). This is particularly critical and becomes problematic for species with small rates of change. And one can certainly expect significant auto-correlation for long-lived gases. Did you account for auto-correlation in your uncertainty estimates reported in Table 5? They appear rather small in both cases (using the Gardiner or Timofeev methods) and your comment on lines 354-355 puzzles me.

Figure 8: the seasonal modulations for the various data sets are presented, and it is immediately obvious that surface measurements are much more flat than the others. But do you think that a direct comparison is meaningful? Unlike remote-sensing

[Figure]

measurements, surface sampling is unaffected by atmospheric dynamics (tropopause height changes, pressure variation...) and the include very high frequency measurements, leading to very robust averages. It might be useful to alert the readers of these specific and different situations.

Typos

Typos were not systematically searched for. I just spotted a few ones. See below. And the list of references was not thoroughly checked.

Line 32: replace FS1125HR by IFS125HR

Line 120: replace FS1125HR by IFS125HR

Line 316: replace HCFH-22 by HCFC-22

Line 319: replace Analisys by Analysis

References

Barthlott, S., Schneider, M., Hase, F., Wiegele, A., Christner, E., González, Y., Blumenstock, T., Dohe, S., García, O. E., Sepúlveda, E., Strong, K., Mendonca, J., Weaver, D., Palm, M., Deutscher, N. M., Warneke, T., Notholt, J., Lejeune, B., Mahieu, E., Jones, N., Griffith, D. W. T., Velazco, V. A., Smale, D., Robinson, J., Kivi, R., Heikkinen, P. and Raffalski, U.: Using xCO2 retrievals for assessing the long-term consistency of NDACC/FTIR data sets, Atmos. Meas. Tech., 8(3), 1555–1573, doi:10.5194/amt-8-1555-2015, 2015.

Bernath, P. F., Steffen, J., Crouse, J. and Boone, C. D.: Sixteen-year trends in atmospheric trace gases from orbit, J. Quant. Spectrosc. Radiat. Transf., 253, 1–19, doi:10.1016/j.jqsrt.2020.107178, 2020. Blumenstock, Th., et al., 10.5194/amt-2020-316, paper in discussion for AMT, 2020. Brown, A. T., Chipperfield, M. P., Boone, C., Wilson, C., Walker, K. A. and Bernath, P. F.: Trends in atmospheric halogen containing gases since 2004, J. Quant. Spectrosc. Radiat. Transf., 112(16), 2552–2566,

doi:10.1016/j.jqsrt.2011.07.005, 2011.

Chirkov, M., Stiller, G. P., Laeng, A., Kellmann, S., von Clarmann, T., Boone, C. D., Elkins, J. W., Engel, A., Glatthor, N., Grabowski, U., Harth, C. M., Kiefer, M., Kolonjari, F., Krummel, P. B., Linden, A., Lunder, C. R., Miller, B. R., Montzka, S. A., Mühle, J., O'Doherty, S., Orphal, J., Prinn, R. G., Toon, G., Vollmer, M. K., Walker, K. A., Weiss, R. F., Wiegele, A. and Young, D.: Global HCFC-22 measurements with MIPAS: retrieval, validation, global distribution and its evolution over 2005–2012, Atmos. Chem. Phys., 16(5), 3345–3368, doi:10.5194/acp-16-3345-2016, 2016.

Montzka, S. A., Dutton, G. S., Yu, P., Ray, E., Portmann, R. W., Daniel, J. S., Kuijpers, L., Hall, B. D., Mondeel, D., Siso, C., Nance, J. D., Rigby, M., Manning, A. J., Hu, L., Moore, F., Miller, B. R. and Elkins, J. W.: An unexpected and persistent increase in global emissions of ozone-depleting CFC-11, Nature, 557(7705), 413–417, doi:10.1038/s41586-018-0106-2, 2018.

Polyakov, A. V., Timofeyev, Y. M., Virolainen, Y. A., Makarova, M. V., Poberovskii, A. V. and Imhasin, H. K.: Ground-Based Measurements of the Total Column of Freons in the Atmosphere near St. Petersburg (2009–2017), Izv. Atmos. Ocean. Phys., 54(5), 487–494, doi:10.1134/S0001433818050109, 2018.

Prignon, M., Chabrillat, S., Minganti, D., O'Doherty, S., Servais, C., Stiller, G., Toon, G. C., Vollmer, M. K. and Mahieu, E.: Improved FTIR retrieval strategy for HCFC-22 (CHClF 2 ), comparisons with in situ and satellite datasets with the support of models, and determination of its long-term trend above Jungfraujoch, Atmos. Chem. Phys., 19(19), 12309–12324, doi:10.5194/acp-19-12309-2019, 2019.

Rigby, M., Park, S., Saito, T., Western, L. M., Redington, A. L., Fang, X., Henne, S., Manning, A. J., Prinn, R. G., Dutton, G. S., Fraser, P. J., Ganesan, A. L., Hall, B. D., Harth, C. M., Kim, J., Kim, K. R., Krummel, P. B., Lee, T., Li, S., Liang, Q., Lunt, M. F., Montzka, S. A., Mühle, J., O'Doherty, S., Park, M. K., Reimann, S., Salameh, P. K., Simmonds, P., Tunnicliffe, R. L., Weiss, R. F., Yokouchi, Y. and Young, D.: Increase

in CFC-11 emissions from eastern China based on atmospheric observations, Nature, 569(7757), 546–550, doi:10.1038/s41586-019-1193-4, 2019.

Santer, B. D., Wigley, T. M. L., Boyle, J. S., Gaffen, D. J., Hnilo, J. J., Nychka, D., Parker, D. E. and Taylor, K. E.: Statistical significance of trends and trend differences in layer-average atmospheric temperature time series, J. Geophys. Res., 105(D6), 7337–7356, doi:10.1029/1999JD901105, 2000.

Steck, T.: Methods for determining regularization for atmospheric retrieval problems, Appl. Opt., 41(9), 1788, doi:10.1364/AO.41.001788, 2002.
* * *

---

## Referee Comment (RC2) · Anonymous Referee #2 · 20 Jan 2021

Review of "Measurements of CFC-11, CFC-12, and HCFC-22 total columns in the atmosphere at the St. Petersburg site in 2009-2019", A Polyakov, et al.

This article presents multi-year trends in CFC11 CFC12 and HCFC22 measured at St. Petersburg. It describes the retrieval of the vertical profiles from solar absorption spectra. Then goes on to analyze the trends and compare with other independent datasets.

While the final results and comparisons are reasonable the description of the spectral analysis and retrieval process is deeply limited and flawed and have be improved before acceptance for publication is conferred. Specific issues related to this are given in the

'Major' section.

Major:

L 106: "The difficulties of the freons TCs retrievals are caused, first of all, by small values and a smoothed spectral dependency of the radiation absorption by these gases which lead to the low information content of the FTIR measurements with respect to the freons TCs." This statement is vague, and poorly worded. While it may be colloquially expressing a practical opinion of someone doing retrievals it could be and would be more useful to readers if filled out more technically.

L128 – 136: "The analysis of the Inverse Problem Solution Process (IPSP)", This apparent procedure is not described, therefor the methodology to determine the characteristics of the QHN, its full effect on spectra and subsequent retrievals, perhaps straight forward or perhaps more sophisticated is unknown. It likely would be of wide interest. Consequently the reader does not know how the author came to the exclusion of some 450 spectra.

L145-150: "the main criterion for choosing the optimal values of setup parameters was the stability of the target gas TCs during a day. More precisely, the root mean square value (RMS) over all days for SD of the gas TCs per a day was minimized. Along with the daily variability of the TCs, the mean value and the SD of the information content of measurements (degrees of freedom for signal, DFS) (Rodgers, 2000, p. 19) as well as the estimates of the systematic and random measurement errors and the spectral residual — the RMS difference between measured and calculated spectra for the retrieved state of the atmosphere ($\chi 2$)" Listing these does not explain how they are used. This section requires a thorough explanation.

L150: "Table 1 presents the main optimized parameters obtained in previous studies." Table 1 does not specify any of the mention retrieval parameters.

L152 – 154: "Target gas absorption is calculated based on pseudo–lines (see

mark4sun.jpl.nasa.gov/pesudo.html for pseudo–lines), interfering gases absorption is calculated based on spectroscopic information from the HITRAN database", this is not true please see the list of interfering species.

L163: "The main factor that determines the shape of the SBL is the filter spectral transmission function (STF)." The spectral baseline is typically 0% transmission line. It typically is not affected by the optical filter transmission or envelope.

L163 – 169: Several points require more detail. Does the water vapor continuum effect the artificial light source spectra? The solar spectra? or both? "contribution in the considered spectral region under conditions of the St. Petersburg site can significantly exceed 50 %." This 50% of what exactly?, "For a 30 cm−1 window, the selectivity of continual uptake is sufficient to influence the IPSP results." Completely unclear what this statement refers to. If this this continuum is a feature of the spectra and well modeled then some plot should be shown to prove it has been resolved.

L170 – 183: This paragraph tries to explain the process of modeling the ice. Its is still not clear where the ice is in the optical path. But the mention of LN2 assumes its at the detector. This should be made clear. Was the WV continuum modeling used in the retrieval? Appears not but its not clear. It appears the a simple quadratic background was used as is standard in many retrievals. The term cryo-sediment does not seem appropriate for the feature.

L205 – 214: The author should explain the large difference in the curves F12 versus F11 & F22

L215-239: and Fig's 2-4: This section seems to compare a single (per species?) constraint called 'OE' with an alpha optimized T-P constraint. First the OE a priori (Sa) is not given or described. Further since it is only one of a large possible array of constraints the comparison is in no way of general significance or value to the reader as well as mis-labeled. This section is so lacking in information as to mis lead the reader. This section needs significant redress before re-submission. To wit the final statement

regarding the contradicting conclusion found in Prignon is not explained.

Table 3 & 4: Table 3 and the upper section of table 4 should be combined into one table and the lower part of table 4 should stand alone as table 4 giving the uncertainties of the retrievals.

L245 – 250: Please add to the tables how many spectra were removed in each step to remove outliers. This would be instructive on 'far' versus 'near' outliers were removed & overall data quality.

L260: "spectral residuals vary from 0.34 to 0.52 % depending on the gas; it corresponds to the SNR values of 209,280, and 327" please explain (equation?) how these correspond?

L265 – 272: Earlier the authors state they have a modeled covariance. This could be a reasonable estimate and consequently the calculation could be performed and would be informative.

L293 – 297: There is no explanation of why or how the optical filter could have such an effect on the variability as it is a static or passive component. Some explanation is required to support this statement.

  Minor:

L57 "Since Molina and Rowland (1974) have reported that CFCs accumulated in the Earth's atmosphere lead to an increased rate of ozone depletion, the attention of both scientists and policymakers to the ozone hole problem has been increasing. " This statement may have been true in the 1990's but not so today.

L66 – Use of atmospheric content is not standard, often atmospheric burden when referring to the total column is used.

L73 & Amendments should be added after Montreal Protocol

L86-95 This review is not thorough. Certainly, any review of FTIR CFC efforts needs to

include Rinsland 2010 and references therein.

L111: "the Tikhonov–Phillips (T–Ph) approach which is more suitable for long–lived gases with a pronounced trend." – this statement is obvious or well known and requires a reference.

L124: "The observational system is based on a Bruker FS125HR Fourier spectrometer, but some of the equipment is non–standard." Doe the author mean in an NDACC-IRWG sense?

L125: "a non–standard spectral filter F3 was used for measurements in the spectral region with considered freons absorption bands." There is no reference for 'F3'. If a local name it should be referenced as such.

L128 – 136: The author should also refer to this as 'channeling' its more common name and insert the in press [Blumenstock AMT 2021] for a reference.

L137: "For a preliminary assessment of the signal to noise ratio (SNR), the standard deviation (SD) of the signal", Its not clear but presumably the SD of the SNR, Please clarify.

L146: "More precisely, the root mean square value (RMS) over all days for SD of the gas TCs per a day was minimized. " This not clear at all, please re-phrase.

L150: chiˆ2 is within the nomenclature of SFIT is a normalized part of the convergence criteria. Is it being used here in that capacity or of simply renaming RM = chiˆ2? This needs clarification.

Table 2: Not readable needs to be reformatted with clear rows and columns

L190: "by a priori information of the Tikhonov–Phillips" T-P is an ad hoc constraint not actually a priori information.

L192: "Unlike the OE, the T–Ph approach does not "pull" the solution to the mean profile", The OE does not pull, the retrieved profile retains the a priori value when there

is no new information from the spectra. Also not to the 'mean' rather the a priori.

L195: "the OE approach requires the use of the covariance matrices for describing the variability of the target gases profiles," not so, an array of ad hoc constrains can be applied within the OE context.

L202: (sp) choice

L209: 'and the both' is awkward maybe should be 'and both'

L211: if the author is referring to a profile scaling procedure it should be clearly stated so e.g. "first guess profile multiplier."

L216: specify section and / or page of appropriate discussion in Rodgers & Connor 2003.

L251: "geographical latitude", is redundant.

L248: does "not provide a solution" mean not converge or other issues, or both?

L310: "which does not have a systematic component during a day, to the random error." This is not clear, What is a systematic component to a random error?

EQ2 sin()

L406: 'belt' might better be 'range'

L418: "a noticeable seasonal variations" rather: "a noticeable seasonal variation"
* * *

---

## Author Comment (AC1) · 9 Mar 2021

Major concerns
As indicated above, the major concern with this contribution is that some of the conclusions are presented as universal truth while they are not backed by the material presented. These conclusions are all related to the choice of a Tikhonov-Phillips (TPh) regularization that would be more appropriate than OE, e.g.:
– T-Ph regularization it is more suitable for long-lived gases with a pronounced trend
– the profiles retrieved by the OE method are less reliable, showing distorted profile shapes when compared to T-Ph results
Although the T-Ph approach and its tuning are discussed in length in the manuscript, important information is missing regarding the OEM retrievals. No details are given (or even summarized) on the choice of the many OEM parameters, on the resulting information content (DFS), on the error budgets. And what about the efforts deployed and the indicators used to validate the definition of the OEM a priori states for the three targets? A paper by the same authors listed in the references (Polyakov at al. , 2018) might provide these details, but unfortunately this work is not freely available to me.
If we miss this kind of information, we cannot be convinced that T-Ph is necessarily a better option than OEM. It might well be that non-optimum OEM results are compared with carefully tuned T-Ph products. Then it is potentially comparing apples and pears in Figures 2 to 4.

*Really, Polyakov et al. (2018) did not analyze the dependence of solution result and parameters on apriori matrix. They used matrices most suitable for a retrieval solution at that moment. We included the description of matrices considered in the text. And, yes, we agree, it is not correct to compare the retrievals with finely optimized T-Ph parameters with that with more general apriori OE matrices. Thus, we cannot make a conclusion on the T-Ph approach advantage. We made changes in a discussion of the figures and the table as well as in the conclusions. But we presume, T-Ph method must be a best solution, therefore, we analyzed it in the manuscript. Optimization of OE matrices can be an interesting problem but it is out of a scope of the manuscript. We included in the text the information about approach used by Polyakov et al. (2018).*

In earlier studies, Polyakov et al. (2018) used the OE method (Rodgers, 2000, p. 65) for solving the inverse problem, i.e. apriori information was given in the form of a mean profile and a covariance matrix with 0.05 relative uncertainty at all altitudes and a correlation coefficient of 5 km built on the basis of the WACCM dataset for the period of 1980-2020.

Regarding these figures, it is also important to keep in mind that total columns are retrieved, or a single piece of information (DFS is merely larger than 1). Then the shape of the retrieved profiles is not very relevant, especially because the computed error bars affecting each single (and meaningless taken alone) mixing ratio are so large that they define a broad range of possible solutions with an identical total column. A profile with a more realistic form could likely be drawn without exceeding the resulting uncertainty ranges.

*A shape of retrieved profiles also made a sense for total columns due the dependence of absorption on temperature and pressure that change with the altitude.*

Another reason given by the authors for selecting T-Ph against OEM regularization is the lack of supporting geophysical information needed to build a covariance matrix: "it should be taken into account that the OE approach requires the use of the covariance matrices for describing the variability of the target gases profiles, preferably the real covariance matrices, that are unavailable for the considered freons". However, 16 years

of ACE profiles are available, they could be used (and consistently extrapolated down to SPB altitude) to construct it, including for the extra diagonal elements. MIPAS data products are also available to the community (see Chirkov et al., 2016). Then at least two multi-year data sets with global coverage.

*These data indeed have a high scientific value, but they cannot be used for constructing covariance matrices for several reasons: 1) The data of limb or solar occultation usually do not allow information on atmospheric gases below 6-7-8 km, whereas the target gases are mainly located in the troposphere. The use of extrapolation to construct covariance matrices to altitudes with the largest content of a gas makes it meaningless to use the measured profiles. 2) The horizontal resolution of these measurements is of hundreds (typically 300-500) km, thus satellite measurements are in a poor agreement with local vertical profiles, for which the covariance matrix is needed. 3) The construction of local matrices for St. Petersburg is complicated by a small number of ACE-FTS measurements for this location and a short period of MIPAS measurements. We indicated these reasons in the text.*

OE approach, with the apriori information given in a form of normal distribution of the target gas profile, requires the use of the covariance matrices for describing the variability of the target gases profiles, preferably the real covariance matrices, that are unavailable for halocarbons considered. Although there are datasets of available satellite measurements (ACE-FTS and MIPAS) of halocarbons, they cannot be used for constructing covariance matrices for several reasons: 1) The data of limb or solar occultation usually do not provide information on atmospheric gases below 7⬚8km, whereas the target gases are mainly located in the troposphere. The use of extrapolation to construct covariance matrices to altitudes with the largest content of a gas makes it meaningless to use the measured profiles. 2) The horizontal resolution of these measurements is of hundreds (typically 300⬚500) km, thus satellite measurements are in a poor agreement with local vertical profiles, for which the covariance matrix is needed. 3) The construction of local matrices for St. Petersburg is complicated due to a small number of ACE-FTS measurements for this location and a short period of MIPAS measurements.

Second, OEM regularization could make use of ad hoc parameters such as to determine the a priori states. This might sound as artificial, but it is not that different than setting up the alpha for T-Ph. It is true however that it could be trickier, because several parameters are available: the per-layer a priori covariance, the type of inter-layer correlation (Gaussian, exponential), the correlation length.

*Yes, indeed, but this choice is difficult to approve. This approach was used by Polyakov et al 2018.*

Note by the way that for long-lived and well-mixed gases, it could be relevant to set up a length of several kilometers in order to avoid oscillations in the retrieved profiles.

*Polyakov et al. (2018) used a value of 5 km for length.*

Finally, it should be noted that the gases involved in the present study do not present "pronounced trends". CFC-11 and CFC-12 will have only changed by about 5% after 10 years. Even if it is larger for HCFC-22 (  25% over 2009-2019), this remains less than the changes observed for some other FTIR products over a single year. For example, ethane presents total columns varying on average by more than a factor two over a season, and still OEM regularization is fully and successfully applicable.

*We used an incorrect word. We should use "reliable trends" instead of "pronounced trends". But it is more important to point out that trend estimates are one of the aims of the measurements analysis. The "dragging in" the retrieval to the mean (apriori) profile for measurements with low information content (DFS<1.5) certainly influence the trend estimate.*
*We remove the phrase and change it to:*
In the current study, we used the Tikhonov–Phillips (T–Ph) approach to improve the retrieval strategy as in contrast to the optimal estimation (OE) method it does not "pull" the solution to the apriori profile for measurements with low information content. Moreover, the T-Ph approach allows more stable results to be derived than the OE method (e.g. Senten et al., 2012).

Addressing the concerns detailed above could in my opinion be done in two ways. First, provide some information on how the OEM regularization was optimized, and supply elements allowing the reader to evaluate and compare on more solid grounds the data products, in terms of uncertainty, daily scatter, DFS: : :
*We pointed the matrices of Polyakov et al (2018) parameters, that allow reader estimation valuable of the comparisons.*

Or to significantly reduce section 2.3 and to give focus on the valuable derived time series and the comparisons with other data sets.
*And we shifted the focus by removing a statement on advantage of T-Ph regularization.*

In any case, one should not be left with the message that T-Ph regularization is necessarily the good option for the retrievals of CFCs or HCFCs. Such
conclusions cannot be reached with a single site study, especially when involving challenging conditions. It remains to be demonstrated that it would also be the case for
other FTIR stations. So adding "for our site", "in our case" at some selected places in
the text could be appropriate to temper the argumentation and make it more specific.
*We agree, this is true only for our site. We did not suggest using the strategy developed directly at other sites. Strategy may be used as a part of optimization and development of the unified retrieval strategy for all sites, for example, as a first guess.*
*We corrected the text defining that the technique developed can be used as a basis for developing the retrieval strategies at other sites, i.e. in the conclusion:*
The retrieval techniques suggested may be used in further development of unified strategies for halocarbons retrieval at the sites of the NDACC observational network that use the Bruker IFS125HR spectrometers for IR solar spectra measurements.

Second order issues and minor comments
I think the one-sentence description (starting line 42) of the impact of halogenated source gases on the formation of the ozone hole is a bit oversimplified and may deserve an additional statement informing about the roles of the stratospheric reservoirs of chlorine and of the heterogeneous reactions in chlorine activation. The gas phase chemistry does not explain the massive polar ozone depletion, and the photolysis of CFCs does not happen in the Polar Regions since there is no high-energy UV photon available there. Another option is to identify and include a good reference.
*We corrected the text:*
Although the major content of these gases is concentrated in the troposphere, in the equatorial region the global circulation moves them out into the lower and middle stratosphere and transports to high–latitude regions. In the stratosphere, CFCs are photochemically decomposed to chlorinated free radicals (Cl, ClO) that are deactivated into chlorine reservoirs HCl, ClONO2,

and HOCl (WMO, 1985, Chapter 3). In polar regions, the heterogeneous reactions on the surfaces of polar stratospheric clouds and cold sulfate aerosols convert inert reservoir molecules into active forms that photolyze producing free radicals and cause the chemical ozone depletion in spring through catalytic cycles resulting up to the appearance of so–called ozone holes (Solomon et al., 2014).

Line 45: the Montreal Protocol does not act directly on the halocarbons emissions. Instead, the production of the relevant gases are limited and then banned. I suggest replacing "emission" by "production" on line 45.
*Done*
As the result of the Montreal Protocol and its amendments and adjustments that restricted the production of chlorofluorocarbons …..

Line 61: this sentence is somewhat misleading: the phase-out (100%) of the CFCs was decided in 1992, for a complete implementation by the end of 1995 (Copenhagen Amendment). In 1989, only a reduction was enforced by the initial treaty.
*Corrected*
The Montreal protocol from 1987, which came into force in 1989, limited production and consumption of CFCs. Later on, in 1992 in Copenhagen and in 1995 in Vienna, phase-out of CFCs was stated by the end of 1995 in developed countries and by the 55 end of 2010 in developing countries.

Line 62: Brown et al. 2011 is a good reference, but more recent trends have been published by the ACE team, considering now 16 years of measurement and improved versions of the data. I strongly suggest considering here and each time it is relevant these updated results (see Bernath et al., 2020), also for the trend comparisons.
*We corrected the text.*
ACE-FTS satellite measurements in last 16 years (Bernath et al., 2020) illustrated the success of the Montreal Protocol by estimated decreasing trends in CFC-11 (-0.53 % per year) and CFC-12 (-0.61 % per year) abundancies and a slowing rate of increase in HCFC-22 abundancies (1.8 % per year).

Lines 63-66: it also depends on the evolution of the bromine and nitrogen stratospheric loadings!
*We corrected the text.*
The time needed for recovery of the ozone layer **among other factors** depends on the sustainability of the reduction in the concentration of CFC-11, CFC-12 and other halocarbons in the atmosphere.

Line 74: to my knowledge, the HFCs are targeted by the Kigali Agreement, not the HCFCs. HCFCs regulation is under the earlier amendments or adjustments. Please check and amend if needed.
*Yes, Corrected*
Under the Montreal Amendment (1997) all countries must gradually phase down HCFCs.
In September 2007 it was decided to accelerate the phase-out of HCFCs. Developed countries had been reducing their consumption of HCFCs and completely phasing them out by 2020. Developing countries agreed to start their phase out process in 2013 and are now following a stepwise reduction until the complete phase-out of HCFCs by 2030.

Line 83: do you mean down to the Earth surface?

*Of course,* down *, corrected*

Line 86: it might be true that the publications on the subject were rather episodic, but not the measurements which were continuously performed and exploited at some of the NDACC sites. For example, halocarbon FTIR time series have been systematically included in the successive editions of the WMO assessments on ozone depletion.
*We took this remark in account, added:*
Time series of CFC-11, CFC-12, and HCFC-22 TCs above Jungfraujoch station, Switzerland, are presented in periodic WMO reports on Scientific Assessment of Ozone Depletion (e.g. WMO, 2018).

Line 89: note that "freon" is a registered trademark.
*We changed the term «freon» by «halocarbon».*

Lines 93-94: this is not correct, Prignon et al. (2019) proposed an approach for the determination of total *and* partial columns, for the 1988-2017 time period (three decades), not only for 1999-2018.
*You are right, the retrieval algorithm allows deriving not only total column of the target gases but partial columns in troposphere and stratosphere as well. This is a misprint, surely, 1988-2017.*
*Corrected.*
Prignon et al. (2019) proposed a technique for estimating two partial columns and TCs of HCFC-22 at the Jungfraujoch mountain station and corresponding time series of HCFC-22 TCs for 1988-2017 along with the trend analysis for various time periods.

Lines 106-108: I somewhat disagree with the arguments on the situation as to the absorptions and spectral signatures (quoted as low and smoothed). In my opinion, this discussion as presented misses the fact that the situation is very different for the three target gases: CFC-11 is likely the more difficult with a broad feature perturbed by strong water vapor lines; CFC-12 has a stronger and more isolated signature peaking at more than 10% and HCFC-22 presents a narrower feature quite free of interferences (HWHM probably on the order of ~0.05 cm-1), resulting in the possibility to select a less wider micro-window. Of course, the spectral scenes will also be influenced by the latitude and altitude of the station.
*Yes, you are right. HCFC-22 is quite narrower. The text is corrected.*
Finally, the CFC-12 and at a larger extent the CFC-11 bands have smoothed spectral dependency of absorption that requires to use wide micro-windows for retrieving their abundancies, 2 cm-1 for CFC-12 and not smaller than 30 cm-11 for CFC-11.

Line 126: I guess QHN is another description of channeling? Note the relevant discussion paper on AMTD by Blumenstock et al. (2020). Harmonizing the designations could be helpful.
*We changed everywhere the term on commonly used, i.e.*
Since this filter was plane–parallel, a parasitic interference arose in it, leading to the appearance of an effect of the optical resonance ("channeling"), see (Blumenstock et al., 2020).
….
The period of the channeling is caused by the material and the thickness of the filter, and in the spectral region 800-900 cm$^{-1}$ it is close to 1.1 cm$^{-1}$, while the amplitude of the channeling varies from zero to a few percent value depending on random filter positioning parameters.
etc

Line 163: the discussion on the spectral transmission function is interesting and original. If I understand correctly, the selection of the relevant parameters is conducted such as to limit the CFC-11 intraday variability. In the end, do you see any correlation between the water vapor and the CFC-11 total columns?

*We added the information on water vapor continuum in the text and the figure in Appendix:* Water vapor continuum makes a significant contribution to radiation attenuation by the atmospheric water vapor (Mlaweret al., 2012). Our calculations have shown that radiation absorption by water vapor continuum in the considered spectral region under conditions of the St. Petersburg site can significantly exceed 50‰. For a 30 cm⁻1 window, the selectivity of 205 continual uptake is sufficient to influence the spectra processing results. To calculate the water vapor continuum, we used a free–distributed computer code (MT_CKD, 2017) and the daily profiles of water vapor independently derived from the FTIR measurements (Virolainen et al., 2017). In the first approximation, the contribution of the water vapor continuum to absorption is proportional to the water vapor partial pressure squared, and it can be detected only in a very humid atmosphere. We estimated the contribution of water vapor continuum numerically by analyzing spectra with and without it considering on the most humid days in 2018: July 29, August 2 and 9. The neglecting continuum absorption in these days led to overestimates of the CFC-11 TCs by an average of 2.9 %. Although this value is less than the measurement error (see Table 4), it systematically depends on the water vapor content. Therefore, it is necessary to consider the water vapor continuum. In Fig. A1 of Appendix A, we presented an example of transmission due to absorption by water vapor continuum

Lines 236-239: the statements regarding the Prignon et al (2019) paper are not correct. These authors also used a T-Ph regularization with alpha=9, minimizing the smoothing and measurement errors as per Steck (2002). Please amend your text accordingly.

*Yes, they used T-Ph approach with alfa parameter equals to 9, minimizing the smoothing error and retrieval errors. We corrected the error in the text.*
Prignon et al. (2019) reported the higher values of DFS (DFS = 1.97) caused by T-Ph regularisation 285 with parameter alpha = 9 and a low atmospheric water vapor content above the mountain (3580 m a.s.l) site Jungfraujoch.

Line 268: regarding the smoothing error, Prignon et al. (2019) have indeed evaluated it to be small (see Table 1 in their paper).

*Thank you, we gave a reference to Prignon et al. (2019), although this estimate was done for the OE method.*
Prignon et al. (2019) showed that the smoothing error for HCFC-22 is rather small (0.3 %).

Line 297: is the mean molar fraction (or MMF) another name for the dry air mole fraction (often denoted xTARGET, see e.g., section 2.4 in Barthlott et al., 2015), as used by the NDACC and TCCON communities? If yes, it might be good here also to harmonize the designations.

*We changed it on the "Xgas" type and added:*
Xgas are calculated by dividing the gas total column by the dry air column.

Table 4 reports much higher daily SD for HCFC-22 than for other gases. Is this a filter 3 effect (before 02/2016)? Or is this just because intraday variability for an unregulated gas might be affected by polluted episodes (excursion above the baseline in in situ surface time series)? Also and with the exception of HCFC-22, the random errors

quoted on line 18 are significantly larger than the intraday SD on line 4. Shouldn't they be commensurate?

This question is discussed in lines 300-308. This is due to a large contribution of the uncertainty of the temperature profile for CFC-11 & CFC-12 and the predominance of the contribution of spectral noise for HCFC-22 to the random error.

Trends on Table 5: it is more and more clear that the uncertainties affecting trends are often underestimated because the methods used do not account for the autocorrelation present in the time series (see e.g., Santer et al., 2000). This is particularly critical and becomes problematic for species with small rates of change. And one can certainly expect significant auto-correlation for long-lived gases. Did you account for auto-correlation in your uncertainty estimates reported in Table 5? They appear rather small in both cases (using the Gardiner or Timofeev methods) and your comment on lines 354-355 puzzles me.

*We corrected the text:*

Gardiner et al. (2008) model the intra-annual variability in terms of a Fourier series, Timofeev et al. (2020) use monthly mean values of the considered period to describe a seasonal cycle. In both methods, trends were estimated by subtracting the seasonal variability from initial time series. In first method, we considered periodicities of 4 month and larger, in second method, monthly mean values accounted for periodicities from 1 month. The estimation of the width of the confidence interval of the trend value for the Gardiner's approach was carried out using the Bootstrap method, for the Timofeev's method, it was calculated on the basis of a theoretical statistical approach. It is worth mentioning that we did not take into account the autocorrelation that could be presented in long-lived Xgas time series. Santer et al. (2000) demonstrated that neglecting of the autocorrelation in time series can affect the trends estimates and underestimate uncertainties, however due to substantially irregular FTIR measurements it was difficult to estimate it.

Figure 8: the seasonal modulations for the various data sets are presented, and it is immediately obvious that surface measurements are much more flat than the others. But do you think that a direct comparison is meaningful? Unlike remote-sensing measurements, surface sampling is unaffected by atmospheric dynamics (tropopause height changes, pressure variation: : :) and the include very high frequency measurements, leading to very robust averages. It might be useful to alert the readers of these specific and different situations.

*We added into the manuscript:*

There are some fundamental differences between local surface and remote sensing measurements (satellite and ground-based FTIR). First, surface measurements are performed with a high regularity and frequency, resulting in stable averages. Secondly, they are unaffected 480 by variations in pressure and tropopause height. And finally, the surface data used were obtained in close to background conditions. Therefore, Figure 8 demonstrates the low seasonal variability of the GVMR – within tenths of a per cent for CFC-11 and CFC-12 and within 0:7% for HCFC-22.

Typos
Typos were not systematically searched for. I just spotted a few ones. See below. And the list of references was not thoroughly checked.
Line 32: replace FS1125HR by IFS125HR
Line 120: replace FS1125HR by IFS125HR
Line 316: replace HCFH-22 by HCFC-22
Line 319: replace Analisys by Analysis

*Done.*

---

## Author Comment (AC2) · 9 Mar 2021

*We thank the referee for valuable and useful comments and remarks.*

While the final results and comparisons are reasonable the description of the spectral analysis and retrieval process is deeply limited and flawed and have be improved before acceptance for publication is conferred.
*We rewrote some parts of these sections, see new version of our manuscript.*

'Major' section

L 106: "The difficulties of the freons TCs retrievals are caused, first of all, by small values and a smoothed spectral dependency of the radiation absorption by these gases which lead to the low information content of the FTIR measurements with respect to the freons TCs." This statement is vague, and poorly worded. While it may be colloquially expressing a practical opinion of someone doing retrievals it could be and would be more useful to readers if filled out more technically.
*We rephrased the text:*
The informativeness of the FTIR spectra with respect to target gases abundancies is low due to several reasons. First, the absorption of target gases is small. Therefore, for example, even for low sun (solar elevation of about 15°), the transmission of solar radiation due to CFC-11 absorption is greater than 90 %, due to HCFC-22 absorption is close to 75 %, and only for CFC-12, the transmission is close to 50 %. Secondly, there is an absorption of interfering gases in the spectral range considered.
 Thus, the CFC-11 absorption band is overlapped with several strong water vapor absorption lines and $HNO_3$ absorption band, each of the CFC-12 and HCFC-22 absorption bands is overlapped with a wing of water vapor absorption line. Finally, the CFC-12 and at a larger extent the CFC-11 bands have smoothed spectral dependency of absorption that requires to use wide micro-windows for retrieving their abundancies, 2 cm$^{-1}$ for CFC-12 and not smaller than 30 cm$^{-1}$ for CFC-11. These factors cause the difficulties in halocarbons retrieval from FTIR spectra measurements.

L128 – 136: "The analysis of the Inverse Problem Solution Process (IPSP)", This apparent procedure is not described, therefor the methodology to determine the characteristics of the QHN, its full effect on spectra and subsequent retrievals, perhaps straight forward or perhaps more sophisticated is unknown. It likely would be of wide interest. Consequently the reader does not know how the author came to the exclusion of some 450 spectra.
*We described in detail the process of preliminary analysis of the spectra quality and slightly corrected the quantities.*
*New text:*
The period of the channeling is caused by the material and the thickness of the filter, and in the spectral region 800-900 cm-1 it is close to 1.1cm$^{-1}$, while the amplitude of the channeling varies from zero to a few percent value depending on random filter positioning parameters. To analyze the presence and the amplitude of the channeling, we performed the Fourier analysis in the most transparent spectral range 892-905cm$^{-1}$ for harmonic components with periods in the intervals of 1-1.25cm$^{-1}$ . The channeling amplitude was calculated relative to the mean signal value in this spectral range.
It is worth mentioning that the SFIT4 software supports the accounting for channeling and its compensation in a spectrum. However, our estimations showed that when the channeling amplitude exceeded 2%, this compensation became unsatisfactory due to a significant increase of scatter in the results of retrievals. Thus, we excluded such spectra from the further processing.
In addition, we analyzed the autocorrelation coefficient of a dark noise in the range of 660-680cm$^{-1}$ (except a slope) and excluded the spectra with the averaged autocorrelation coefficient greater than 0.1. A large autocorrelation coefficient of the dark noise indicates the presence of

external influences on a measurement process. Moreover, we excluded from the further processing the spectra that were measured at the time when a haze or cloudiness was observed in any part of the sky, since the use of these spectra also noticeably increases the scatter of the results.

As a result of the described filtering, 2901 from 3523 (i.e. 82%) spectra measured before February 2016 were selected for further processing. In February 2016, the F3 filter was replaced by the standard IRWG NDACC filter f6 which wedge–shaped design eliminates the channeling. Thus, the quality of measurements was improved, and 1903 from 1958 spectra were selected, giving in a sum 4804 spectra for 2009-2019 period.

L145-150: "the main criterion for choosing the optimal values of setup parameters was the stability of the target gas TCs during a day. More precisely, the root mean square value (RMS) over all days for SD of the gas TCs per a day was minimized. Along with the daily variability of the TCs, the mean value and the SD of the information content of measurements (degrees of freedom for signal, DFS) (Rodgers, 2000, p. 19) as well as the estimates of the systematic and random measurement errors and the spectral residual âˇAˇT  the RMS difference between measured and calculated spectra for the retrieved state of the atmosphere (_2)" Listing these does not explain how they are used. This section requires a thorough explanation.

*We extended the discussion giving clarification of the criteria used by Polyakov et al., 2019a,b, 2020b :*

In previous studies, Polyakov et al. (2019a, b, 2020a) determined a number of parameters of the retrieval strategy using the SFIT4 code for deriving the TCs of the target gases from the FTIR measurements at the St. Petersburg site: the boundaries of microwindows, the mean/apriori profiles of the measured gases, the magnitude and variability of the zero level, periods for taking into account (or excluding) the channeling, and the background shape of a spectrum (BSS). The criteria used for optimization of retrieval parameters are briefly described below. As we mentioned above, the lifetime of the target gases in the atmosphere is more than 10 years. In addition, CFC-11 and CFC-12 have no known active sources of emission. Therefore, we expect the stability of their retrieved columns during both the each day and the whole period of measurements (excluding trend). At a lesser extent due to its continuous production, the same criterion is valid for HCFC-22, at least for intraday variability. Thus (Polyakov et al., 2019a, b, 2020a) used the stability of the retrieved total columns in terms of minimal root-mean-squared (RMS) SD of the TCs for all days of measurements as the main criterion in selecting the retrieval parameters. Another important retrieval parameter is the number of degrees of freedom for signal (DFS) (Rodgers, 2000, p. 19)

for target gases. As a criterion for optimization, the SD of DFS was minimized. The estimate of total systematic and random measurements errors was also considered. Finally, the residuals (differences between spectra measured and calculated with the retrieved atmospheric state), for estimating of which the RMS residuals normalized to the unit, calculated in the SFIT4 software, and denoted as $\chi 2$, were analyzed. It should be noted that without additional analysis, the listed criteria do not unambiguously determine the optimality of the retrieval technique. The estimate of the TCs measurement errors can be considered as one of the main criteria, but the adequacy of the measurement model also should be taken into account. Thus, for example, by including during spectra analysis the additional unknown parameter as channeling, we increase the measurements errors, however, if we exclude it and residuals get larger, it will indicate that the parameters used are inadequate for real measurements, i. e. the actual presence of channeling in the spectra.

L150: "Table 1 presents the main optimized parameters obtained in previous studies."
Table 1 does not specify any of the mention retrieval parameters.

*The retrieval parameters corresponding to the criteria optimized (lines 145-150) are given in Table 1, as it is defined above.*

L152 – 154: "Target gas absorption is calculated based on pseudo–lines (see mark4sun.jpl.nasa.gov/pesudo.html for pseudo–lines), interfering gases absorption is calculated based on spectroscopic information from the HITRAN database", this is not true please see the list of interfering species.
*We corrected the text.*
Target gas and $COCl_2$ absorption is calculated based on pseudo–lines (see mark4sun.jpl.nasa.gov/pseudo.html for pseudo–lines), other interfering gases absorption is calculated based on spectroscopic information from the HITRAN database which is described in detail by Polyakov et al. (2019a, b, 2020a).

L163: "The main factor that determines the shape of the SBL is the filter spectral transmission function (STF)." The spectral baseline is typically 0% transmission line. It typically is not affected by the optical filter transmission or envelope.
*We used a term SBL for denoting the spectra envelope corresponding to the unit transmission. We corrected the text avoiding using the term SBL.*

L163 – 169: Several points require more detail. Does the water vapor continuum effect the artificial light source spectra? The solar spectra? or both? "contribution in the considered spectral region under conditions of the St. Petersburg site can significantly exceed 50 %." This 50% of what exactly?, "For a 30 cm-1 window, the selectivity of continual uptake is sufficient to influence the IPSP results." Completely unclear what this statement refers to. If this this continuum is a feature of the spectra and well modeled then some plot should be shown to prove it has been resolved.
*The continuum absorption of radiation, in this case, by water vapor in the transparency window of the spectra, is observed in a very humid atmosphere (see Mlawer, et al, 2012). In the Fig. A1 of Appendix A, we presented an example of transmission due to absorption by water vapor continuum for a day with a large water vapor content. In dry atmosphere, this absorption can be neglected. Moreover, we estimated the contribution of water vapor continuum numerically by analyzing spectra with and without it considering in the most humid days and gave the results in the text. The ignoring of continuum absorption in these days led to overestimates of the CFC-11 TCs by an average of 3%. Although this value is smaller than the measurement error, it is systematic and depends on the water vapor content. Therefore, it is necessary to consider the water vapor continuum. All these considerations are valid only for observational sites near the sea level and cannot be applied to the measurements at high altitude sites (like Jungraujoch).*
*New text:*
Water vapor continuum makes a significant contribution to radiation attenuation by the atmospheric water vapor (Mlawer et al., 2012). Our calculations have shown that radiation absorption by water vapor continuum in the considered spectral region under conditions of the St. Petersburg site can significantly exceed 50%. For a 30 $cm^{-1}$ window, the selectivity of continual uptake is sufficient to influence the spectra processing results. To calculate the water vapor continuum, we used a free–distributed computer code (MT_CKD, 2017) and the daily profiles of water vapor independently derived from the FTIR measurements (Virolainen et al., 2017). In the first approximation, the contribution of the water vapor continuum to absorption is proportional to the water vapor partial pressure squared, and it can be detected only in a very humid atmosphere. We estimated
the contribution of water vapor continuum numerically by analyzing spectra with and without it considering on the most humid days in 2018: July 29, August 2 and 9. The neglecting continuum absorption in these days led to overestimates of the CFC-11 TCs by an average of 2.9 %.

Although this value is less than the measurement error (see Table 4), it systematically depends on the water vapor content. Therefore, it is necessary to consider the water vapor continuum. In Fig. A1 of Appendix A, we present an example of transmission due to absorption by water vapor continuum.

Appendix A: Water vapor continuum
Figure A1 depicts the transmission functions of water vapor continuum. Although sometimes a value of precipitable water (PW) above St.Patersburg reaches 50mm, in clear-sky days, when FTIR measurements are performed, maximum value of PW is about 30-40mm.

[Figure]

Figure A1. Transmission function of water vapor continuum for July 29, 2018 - one of the humid days of measurements in 2018 (PW totals 30 mm).

L170 – 183: This paragraph tries to explain the process of modeling the ice. Its is still not clear where the ice is in the optical path. But the mention of LN2 assumes its at the detector. This should be made clear. Was the WV continuum modeling used in the retrieval? Appears not but its not clear. It appears the a simple quadratic background was used as is standard in many retrievals. The term cryo-sediment does not seem appropriate for the feature.

*The ice is located directly on the cooled receiver (HgCdTe detector). The measurements of radiation absorption by ice were carried out using an artificial source of light, therefore the continuous absorption by atmospheric water vapor is not involved in these measurements. The quadratic background shape of the spectra can, of course, be used to simulate various phenomena, but its use must be justified and limited in accordance with the physical nature of the simulated phenomenon. The fact is that the quadratic component of the SBL in the case of a smoothed spectral dependence, like CFC-11, is poorly separated from the gas content, therefore, additional "freedom" (a priori uncertainty of the coefficient) can significantly distort the results obtained.*
*We replaced the term "cryo-sediment" with the term "amorphous water ice" – AWI.*
*New text:*

Repeated measurements of the STF showed that over time they exhibit a specific spectrum of absorption by amorphous water ice (AWI) formed on the HgCdTe detector at the temperatures that has a detector cooled by liquid nitrogen (e.g. Hudgins, et al., 1993; Lynch, 2006). The absorption of radiation by AWI depends on its thickness which increases during the measurement period and decreases during the period of inactivity of the instrument when the receiver is not cooled. In addition, the water vapor from the atmospheric air gradually (on a monthly scale) seeps into the evacuated zone of the instrument and also leads to an increase of the AWI thickness. To compensate for its variability, the BSS was refined with the second–degree wavenumber polynomial implemented in the SFIT4 code. With turning on one more variable, the correction of the BSS curvature specified by the coefficient at the second

power of the wavenumber (hereinafter – curvature value) can lead to "overfreedom" of the solution. To avoid this, the a priori curvature value uncertainty was limited. The parameters for compensating the BSS due to absorption by the AWI were selected in two steps. We minimized the intraday variability of the CFC-11 TCs in a series of
spectra processing and, on the first step, got the a priori thickness of the AWI (0.3 μm for F3, 0.9 μm for f6 filter) with the a priori curvature value of 0. On the next step, we optimized the value of a priori curvature uncertainty as $10^{-6}$ for both filters.

L205 – 214: The author should explain the large difference in the curves F12 versus F11 & F22
*We added the explanation in the text:*
It is worth mentioning that the difference between the curves for different halocarbons in Fig. 1 is meaningful. The presence of a pronounced minimum
for CFC-12 is due to a larger information content of spectral measurements with respect to CFC-12 abundancies compared to CFC-11 and HCFC-22 (DFS for CFC-12 is 1.2, CFC-11 - 1.05, HCFC-22 - 1.0, see Table 3). The reason for this is a weak absorption of interfering gases in the spectral range for CFC-12 retrievals. Thus, for CFC-12, an increase of the regularization parameter, which tightens the requirement for spectrum smoothness, leads to the suppression of useful information on the elements of the vertical gas distribution which is contained in a spectrum. Consequently, the intraday variability of retrievals is
increasing. For CFC-11 and HCFC-22, the informativeness of spectral measurements is less, DFS is close to 1, therefore, large values of the parameter
 and the corresponding requirements for smoothness do not contradict the information contained in a spectrum.

L211: if the author is referring to a profile scaling procedure it should be clearly stated so e.g. "first guess profile multiplier."

*Corrected:*
This can be interpreted as the complete absence of the information on the vertical profile of HCFC-22 in spectral measurements, i.e. only the information on the first guess profile multiplier (profile scaling approach). We performed the retrieval for both the profile scaling and the T–Ph approach with ….

L215-239: and Fig's 2-4: This section seems to compare a single (per species?) constraint called 'OE' with an alpha optimized T-P constraint. First the OE a priori (Sa) is not given or described. Further since it is only one of a large possible array of constraints the comparison is in no way of general significance or value to the reader as well as mis-labeled. This section is so lacking in information as to mis lead the reader. This section needs significant redress before re-submission. To wit the final statement regarding the contradicting conclusion found in Prignon is not explained.

*We rewrote this section:*
It should be noted that the target parameter of the retrieval, TC, is calculated from the initially retrieved vertical profile of the gas. Therefore, it is important to control the retrieval of trace gases profiles. For all three target gases, figures 2–4 depict the sensitivity functions of the TCs to relative variations in the gases profile at different heights (left) (see about averaging kernel (AK) area (Rodgers and Connor, 2003, section 2.1)) and the examples of initial (first guess) and retrieved volume mixing ratio (VMR) profiles (right). All curves are shown for two typical measurements: in a fall–winter season with a low Sun elevation and a low humidity, and in summer, with a high for the site latitude Sun elevation and a wet atmosphere. All parameters in

the figures are given for a regularization of both the OE and the T–Ph methods. Although the T-Ph regularization parameter was
optimized, the covariance matrices of the OE method were taken from (Polyakov et al., 2018), where they were selected from general considerations.

Figures 2–4 demonstrate that the sensitivity, which for the ideal case should be equal to 1 at all heights, turns out to be noticeably lower (from 0.5 to 0.8 for different gases, seasons and methods) at the surface. Then sensitivities increase, reaching a maximum at heights of 8–12 km for CFC-11 and CFC-12, and at heights above 12 km for HCFC-22 which is due to a higher stratospheric content of HCFC-22. Above, the sensitivity decreases which can no longer be significant due to the fall of the VMR of the target gases. As seen from Fig. 2–4, the measurement conditions have a significant effect on the sensitivity functions. In winter, when the Sun elevation is low corresponding to a thicker atmosphere in the solar beam path and low water vapor content, the information content of measurements is higher than in summer. For all three gases, the sensitivity is far from the unit at a greater extent in the lower troposphere in high humidity conditions in summer. Using T–Ph approach and choosing the regularization parameter based on minimizing the intraday variability of TCs, we obtained DFS = 1 for HCFC-22; the DFS value of other two gases is close to 1 (1.05 and 1.20, see Table 3). Prignon et al. (2019) reported the higher values of DFS (D
= 1.97) caused by T-Ph regularisation with parameter $\alpha = 9$ and a low atmospheric water vapor content above the mountain (3580 m a.s.l) site Jungfraujoch.

Table 3 & 4: Table 3 and the upper section of table 4 should be combined into one table and the lower part of table 4 should stand alone as table 4 giving the uncertainties of the retrievals
*Done*

L245 – 250: Please add to the tables how many spectra were removed in each step to remove outliers. This would be instructive on 'far' versus 'near' outliers were removed & overall data quality.
*New table was added as appendix B:*
Table B1. The criteria used and the percentage of data discarded after their application.

| Criterion | CFC-11 | | CFC-12 | | HCFC-22 | |
|---|---|---|---|---|---|---|
| | Value | Excluded, % | Value | Excluded, % | Value | Excluded,% |
| Sys err | 7.96 | 2.9 | 2.58 | 2.5 | 5.91 | 0 |
| Ran err | 4.77 | 5.0 | 4.42 | 4.7 | 9.50 | 4.8 |
| $\chi^2$ | 1.45 | 0.5 | 1.54 | 0.3 | 0.97 | 4.1 |
| DFS | 0.89 | 0 | 1.10 | 1.3 | 0.90 | 0 |
| S/N | 50-600 | 7.9 | 50-600 | 3.7 | 60-600 | 4.2 |
| Not conv | Yes | 0 | Yes | 8.1 | Yes | 3.7 |
| Div | Yes | 3.8 | Yes | 0 | Yes | 3.0 |
| No result | No files | 4.2 | No files | 1.2 | No files | 0 |
| Total excluded | 19% | | 18% | | 16% | |
| Spectra/Days before filtering | 4773 / 720 | | 4768 / 718 | | 4585 / 714 | |
| after filtering | 3864 / 678 | | 3912 / 664 | | 2855 / 663 | |

Rows of Table B1:
1) systematic error (mean plus 2 SD)
2) random error (mean plus 2 SD)

3) residual (Xi2) (mean plus 2 SD)
4) DFS (mean minus 2 SD)
5) To exclude noisy spectra and possible non–linearity in measurements, we use only measurements with SNR values ranging from 50 (60) to 600
6) Not converged
7) Divergence warning
8) SFIT did not present results

L260: "spectral residuals vary from 0.34 to 0.52 % depending on the gas; it corresponds to the SNR values of 209,280, and 327" please explain (equation?) how these correspond?

*We corrected the text, adding word "mean". Since the spectrum in residual calculations is normalized to unit, SNR and residual there are reciprocal values, SNR = 1 / residual. As we are talking about mean values, this equation is approximate.*

*New text:*

Ideally, the spectral residual should be equal to the measurements noise level. For target gases, the mean values of the spectral residuals vary from 0:34 to 0:52% depending on the gas; it corresponds to the SNR values of 209; 280, and 327 for CFC-11, CFC-12, and HCFC-22, respectively. Since the spectrum in residual calculations is normalized to unit, SNR and residual there are reciprocal values, SNR = 1 / residual . Comparing these values to the preliminary determined mean SNR in the opaque spectral range (364;351 and 324, in the same order of gases), we see that for CFC-11 and CFC-12 they are slightly less and for HCFC-22 are nearly the same. This means that for CFC-11 and CFC-12 the radiative transfer model and a set of parameters used, although satisfactory, but not ideally describe the absorption of radiation by the atmosphere and the observational system, whereas for HCFC-22 the retrieval technique works in the best way.

L265 – 272: Earlier the authors state they have a modeled covariance. This could be a reasonable estimate and consequently the calculation could be performed and would be informative.

*We do not have true covariance matrices available. The calculation based on the model matrix depends significantly on this matrix.*

L293 – 297: There is no explanation of why or how the optical filter could have such an effect on the variability as it is a static or passive component. Some explanation is required to support this statement.

*We added an explanation in the text:*

Due to channeling, the F3 filter (used before February 2016), leads to a large scatter in retrieval results owing to larger errors (see section 2.1).

L310: "which does not have a systematic component during a day, to the random error." This is not clear, What is a systematic component to a random error?

*Apparently, the reviewer kept in mind line 301.*

*We removed the phrase below:*

Minor

L57 "Since Molina and Rowland (1974) have reported that CFCs accumulated in the Earth's atmosphere lead to an increased rate of ozone depletion, the attention of both scientists and policymakers to the ozone hole problem has been increasing. " This statement may have been true in the 1990's but not so today.
*Corrected*
After Molina and Rowland (1974) reported that CFCs accumulated in the Earth's atmosphere led to an increased rate of ozone depletion, the attention of both scientists and policymakers to the ozone hole problem had been increased.

L66 – Use of atmospheric content is not standard, often atmospheric burden when referring to the total column is used.
*Corrected*
Based on the 2015–2017 data, Montzka et al. (2018) showed that the rate of change in the CFC-11 atmospheric concentration decreased by …..

L73 & Amendments should be added after Montreal Protocol
Added
In the 2000s, the production and consumption of HCFCs in developed countries have decreased as a response to the Montreal Protocol and its amendments and adjustments.

L86-95 This review is not thorough. Certainly, any review of FTIR CFC efforts needs to include Rinsland 2010 and references therein.

*We studied the papers suggested and added them in overview and in reference sections. Thank you for the useful information.*
First FTIR measurements of atmospheric HCFC-22 were performed from the balloon in early 80-s (Goldman et al., 1981). Spectral resolution of these measurements did not allow halocarbons to be measured from the surface. Later, with the appearance of high-resolution instruments, halocarbons started to be derived with ground-based FTIR spectrometers. In last decades, TCs of halocarbons are measured by ground-based FTIR method more actively (e.g. Notholt, 1994; Rinsland et al., 2005, 2010; Zander et al., 2005; Mahieu et al., 2010, 2013, 2017; Zhou et al., 2016; Prignon et al., 2019).

L111: "the Tikhonov–Phillips (T–Ph) approach which is more suitable for long–lived gases with a pronounced trend." – this statement is obvious or well known and requires a reference.

*We corrected the text and added the following reference.*
Tikhonov–Phillips (T–Ph) approach which allows more stable results to be derived than the optimal estimation (OE) method (e.g. Senten et al., 2012).

C. Senten, M. De Mazi`ere, G. Vanhaelewyn, and C. Vigouroux Information operator approach applied to the retrieval of the vertical distribution of atmospheric constituents from ground-based high-resolution FTIR measurements, Atmos. Meas. Tech., 5, 161–180, 2012, doi:10.5194/amt-5-161-2012

L124: "The observational system is based on a Bruker FS125HR Fourier spectrometer, but some of the equipment is non–standard." Doe the author mean in an NDACC-IRWG sense?

We explained it in the text:

The observational system is based on a Bruker IFS125HR Fourier spectrometer, but some of the equipment is non–standard. In particular, before February 2016 a non–standard (for the IRWG-NDACC sites) spectral filter (hereinafter F3) was used for measurements in the spectral region with target gases absorption bands. Since this filter was plane–parallel, a parasitic interference arose in it, leading to the appearance of an effect of the optical resonance ("channeling"), see (Blumenstock et al., 2020). Moreover, a home-made solar tracking system is used.

L125: "a non–standard spectral filter F3 was used for measurements in the spectral region with considered freons absorption bands." There is no reference for 'F3'. If a local name it should be referenced as such.
*Yes, this notation is now introduced, see previous response.*

L128 – 136: The author should also refer to this as 'channeling' its more common name and insert the in press [Blumenstock AMT 2021] for a reference.
*Corrected, see previous response.*

L137: "For a preliminary assessment of the signal to noise ratio (SNR), the standard deviation (SD) of the signal", Its not clear but presumably the SD of the SNR, Please clarify.

*No, as it is indicated in the text, the SD of the signal is considered, not of the SNR.*

L146: "More precisely, the root mean square value (RMS) over all days for SD of the gas TCs per a day was minimized. " This not clear at all, please re-phrase.

*We corrected the text.*
At a lesser extent due to its continuous production, the same criterion is valid for HCFC-22, at least for intraday variability. Thus (Polyakov et al., 2019a, b, 2020a) used the stability of the retrieved total columns in terms of minimal root-mean-squared (RMS) SD of the TCs for all days of measurements as the main criterion in selecting the retrieval parameters

L150: chi^2 is within the nomenclature of SFIT is a normalized part of the convergence criteria. Is it being used here in that capacity or of simply renaming RM = chi^2? This needs clarification.
Here chi^2 indicates that this is the SFIT estimated value, the normalized residual.
Considered:
the residuals (differences between spectra measured and calculated with the retrieved atmospheric state), for estimating of which the RMS residuals normalized to the unit, calculated in the SFIT4 software, and denoted as $\chi^2$, were analyzed.

Table 2: Not readable needs to be reformatted with clear rows and columns
*We removed the table and rewrote text:*
The using of a wide spectral window for CFC-11 retrieval (30 cm$^{-1}$ , see Table 1) is unusual for deriving the information on the gas content from the high resolution IR spectra and requires the non-standard approach for considering the base spectra shape (BSS). This approach was described in detail by Polyakov et al. (2020a); the main features of this approach are listed below.

The constant and important factor that determines the BSS is the filter spectral transmission function (STF). We have measured the STF in a special experiment using an artificial light source.

Repeated measurements of the STF showed that over time they exhibit a specific spectrum of absorption by amorphous water ice (AWI) formed on the HgCdTe detector at the temperatures that has a detectorcooled by liquid nitrogen (e.g. Hudgins, et al., 1993; Lynch, 2006). The absorption of radiation by AWI depends on its thickness which increases during the easurement period and decreases during the period of inactivity of the instrument when the receiver is not cooled. In addition, the water vapor from the atmospheric air gradually (on a monthly scale) seeps into the evacuated zone of the instrument and also leads to an increase of the AWI thickness. To compensate for its variability, the BSS was refined with the second–degree wavenumber polynomial implemented in the SFIT4 code. With turning on one more variable, the correction of the BSS curvature specified by the coefficient at the second power of the wavenumber (hereinafter – curvature value) can lead to "overfreedom" of the solution. To avoid this, the apriori curvature value uncertainty was limited. The parameters for compensating the BSS due to absorption by the AWI were selected in two steps. We minimized the intraday variability of the CFC-11 TCs in a series of spectra processing and, on the first step, got the apriori thickness of the AWI (0:3 μm for F3, 0:9 μm for f6 filter) with the apriori curvature value of 0. On the next step, we optimized the value of apriori curvature uncertainty as $10^{-6}$ for both filters.

Water vapor continuum makes a significant contribution to radiation attenuation by the atmospheric water vapor (Mlaweret al., 2012). Our calculations have shown that radiation absorption by water vapor continuum in the considered spectral region under conditions of the St. Petersburg site can significantly exceed 50%. For a 30 $cm^{-1}$ window, the selectivity of continual uptake is sufficient to influence the spectra processing results. To calculate the water vapor continuum, we used a free–distributed computer code (MT_CKD, 2017) and the daily profiles of water vapor independently derived from the FTIR measurements (Virolainen et al., 2017). In the first approximation, the contribution of the water vapor continuum to absorption is proportional to the water vapor partial pressure squared, and it can be detected only in a very humid atmosphere. We estimated the contribution of water vapor continuum numerically by analyzing spectra with and without it considering on the most humid days in 2018: July 29, August 2 and 9. The neglecting continuum absorption in these days led to overestimates of the CFC-11 TCs by an average of 2.9 %. Although this value is less than the measurement error (see Table 4), it systematically depends on the water vapor content. Therefore, it is necessary to consider the water vapor continuum. In Fig. A1 of Appendix A, we present an example of transmission due to absorption by water vapor continuum.

Thus for CFC-11 processing, we took into account STF, AWI variability and water vapor continuum.

L190: "by a priori information of the Tikhonov–Phillips" T-P is an ad hoc constraint not actually a priori information.
*Corrected by removing this expression.*

L192: "Unlike the OE, the T–Ph approach does not "pull" the solution to the mean profile", The OE does not pull, the retrieved profile retains the a priori value when there is no new information from the spectra. Also not to the 'mean' rather the a priori.
*When there is no information on target gas in spectral measurements, the retrieval surely equals apriori mean profile. If there is a little information, the solution pulls the apriori mean. The Gaussian distribution is described by the mean and the covariance matrix; in this study, we use an OE with a priori Gaussian distribution of the retrieved vector. We added the word apriori.*

L195: "the OE approach requires the use of the covariance matrices for describing the variability of the target gases profiles," not so, an array of ad hoc constrains can be applied within the OE context.

*We corrected the formulation of the method used.*

…OE approach, with the apriori information given in a form of normal distribution of the target gas profile, requires the use of the covariance matrices...

L202: choice (sp)

*corrected*

L209: 'and the both' is awkward maybe should be 'and both'

*corrected*

L211: if the author is referring to a profile scaling procedure it should be clearly stated so e.g. "first guess profile multiplier."

*corrected*

This can be interpreted as the complete absence of the information on the vertical profile of HCFC-22 in spectral measurements, i.e. only the information on the first guess profile multiplier (profile scaling approach).

L216: specify section and / or page of appropriate discussion in Rodgers & Connor 2003.

*corrected*

Section 2.1

L251: "geographical latitude", is redundant.*corrected*

L248: does "not provide a solution" mean not converge or other issues, or both?

*Both. Either a logical parameter indicates the absence of convergence or divergence of iteration (when the number of iterations reaches the maximum), or there is no file with the results (the software processing is interrupted for unknown reasons).*

L310: "which does not have a systematic component during a day, to the random error."
This is not clear, What is a systematic component to a random error?

*Apparently, the reviewer kept line 301 in mind.*

*In intraday measurements, there is a shift in the retrievals in this day (the reasons may be different). The magnitude of this shift is different for different days. Therefore, when considering the whole data set, this shift is a component of a random error, but when considering one day of measurements, this is a component of a systematic error. We corrected the text.*

EQ2 sin ()

corrected

L406: 'belt' might better be 'range'

*corrected*

L418: "a noticeable seasonal variations" rather: "a noticeable seasonal variation"

*corrected*

---

## Referee Report (RR1)

2nd Review of "Measurements of CFC-11, CFC-12, and HCFC-22 total columns in the atmosphere at the St. Petersburg site in 2009-2019", A Polyakov, et al.

Comments to the authors:

L34, this work does not 'control' implementation …

L38 change to …'suggests optimized'…

L48 change to …qualitatively…

L63 …transports them to…

L88 referring to a phenomenon as 'so-called' connotes that to some degree, the term is inaccurate. In this paper its simply better to be accurate. This should be re-written.

L115 'Until recently… ' is not true or clear. Please see early papers by Rinsland for data back to the 1970's

L125 the statement 'Spectral resolution of these measurements did not allow halocarbons to be measured from the surface'. Is flatly false.

L153 change informativeness to "information content"

L155 while using relative transmission values may be accurate its more customary and informative to use relative absorption when discussing absorption features – this should be changed. Further these absorptions of ~ 10, 25 and 50% are not 'small'.

L196 was the unsatisfactory channel modeling really due to the scatter in the retrievals?

L278 "considering on the most" changed to "considering the most"

L280 "The neglecting continuum" change to "The neglection of the continuum"

L165+ Major: The choice of state vector constraint is fine, the description of why and statements made to contrast with a fictitious 'OE' constraint is false and would easily mislead a reader. Further why no comparisons with a profile scaling retrieval? It's perfectly stable especially for low information content features. This point continues to be a major flaw of the text.

About L: 300- Fairly Major: this discussion is mostly irrelevant. The climatology of these species is not complicated. It is reasonable enough to construct a constraint with WACCM apriori data. In fact, they are computed for SFIT and readily available.

Major: Section at 271: This discussion is interesting but inadequate. The authors only state a difference in TC from a couple tests and do not show or prove it. The spectra in the appendix do not in any way show a difference in the column amount. These slowly varying curves should have an effect on the broad region not dissimilar to the optical filter envelope – although these can change day to day. But the solar viewing instrument uses relative absorption so the absolute transmission is not in general, a concern. That is not

to say the continuum has no effect, rather as the author assumes, it may well have an effect.  What is required here is demonstrated proof.

This is especially true if the authors wish to make a statement that the technique be widely adopted in the NDACC as they do in the conclusions.

This reviewer has now reviewed the document twice.  The document has improved in the second version.  The tables are improved the discussion of the trends is good.  Still, in this second version, there are major issues the need to be corrected.  There are too many minor grammar mistakes to list.  The document requires a complete review as to wording and grammar.  The value, efficacy and methodology of the continuum accounting needs to be described.   The discussion of constraint needs to be more accurate.

Review of "Measurements of CFC-11, CFC-12, and HCFC-22 total columns in the atmosphere at the St. Petersburg site in 2009-2019", A Polyakov, et al.

This article presents multi-year trends in CFC11 CFC12 and HCFC22 measured at St. Petersburg. It describes the retrieval of the vertical profiles from solar absorption spectra. Then goes on to analyze the trends and compare with other independent datasets.

While the final results and comparisons are reasonable the description of the spectral analysis and retrieval process is deeply limited and flawed and have be improved before acceptance for publication is conferred. Specific issues related to this are given in the 'Major' section.

**Major:**

L 106: "The difficulties of the freons TCs retrievals are caused, first of all, by small values and a smoothed spectral dependency of the radiation absorption by these gases which lead to the low information content of the FTIR measurements with respect to the freons TCs." This statement is vague, and poorly worded. While it may be colloquially expressing a practical opinion of someone doing retrievals it could be and would be more useful to readers if filled out more technically.

L128 – 136: "The analysis of the Inverse Problem Solution Process (IPSP)", This apparent procedure is not described, therefor the methodology to determine the characteristics of the QHN, its full effect on spectra and subsequent retrievals, perhaps straight forward or perhaps more sophisticated is unknown. It likely would be of wide interest. Consequently, the reader does not know how the author came to the exclusion of some 450 spectra.

L145-150: "the main criterion for choosing the optimal values of setup parameters was the stability of the target gas TCs during a day. More precisely, the root mean square value (RMS) over all days for SD of the gas TCs per a day was minimized. Along with the daily variability of the TCs, the mean value and the SD of the information content of measurements (degrees of freedom for signal, DFS) (Rodgers, 2000, p. 19) as well as the estimates of the systematic and random measurement errors and the spectral residual — the RMS difference between measured and calculated spectra for the retrieved state of the atmosphere ($\chi 2$)" Listing these does not explain how they are used. This section requires a thorough explanation.

L150: "Table 1 presents the main optimized parameters obtained in previous studies." Table 1 does not specify any of the mention retrieval parameters.

L152 – 154: "Target gas absorption is calculated based on pseudo–lines (see mark4sun.jpl.nasa.gov/pesudo.html for pseudo–lines), interfering gases absorption is calculated based on spectroscopic information from the HITRAN database", this is not true please see the list of interfering species.

L163: "The main factor that determines the shape of the SBL is the filter spectral transmission function (STF)." The spectral baseline is typically 0% transmission line. It typically is not affected by the optical filter transmission or envelope.

L163 – 169: Several points require more detail. Does the water vapor continuum effect the artificial light source spectra? The solar spectra? or both? "contribution in the considered spectral region under conditions of the St. Petersburg site can significantly exceed 50 %." This 50% of what exactly? "For a 30 cm−1 window, the selectivity of continual uptake is sufficient to influence the IPSP results." Completely unclear what this statement refers to. If this this continuum is a feature of the spectra and well modeled then some plot should be shown to prove it has been resolved.

L170 – 183: This paragraph tries to explain the process of modeling the ice. It is still not clear where the ice is in the optical path. But the mention of LN2 assumes it's at the detector. This should be made clear. Was the WV continuum modeling used in the retrieval? Appears not but its not clear. It appears a simple quadratic background was used as is standard in many retrievals. The term cryo-sediment does not seem appropriate for the feature.

L205 – 214: The author should explain the large difference in the curves F12 versus F11 & F22

L215-239: and Fig's 2-4: This section seems to compare a single (per species?) constraint called 'OE' with an alpha optimized T-P constraint. First the OE a priori (Sa) is not given or described. Further since it is only one of a large possible array of constraints the comparison is in no way of general significance or value to the reader as well as mis-labeled. This section is so lacking in information as to mis lead the reader. This section needs significant redress before re-submission. To wit the final statement regarding the contradicting conclusion found in Prignon is not explained.

Table 3 & 4: Table 3 and the upper section of table 4 should be combined into one table and the lower part of table 4 should stand alone as table 4 giving the uncertainties of the retrievals.

L245 – 250: Please add to the tables how many spectra were removed in each step to remove outliers. This would be instructive on 'far' versus 'near' outliers were removed & overall data quality.

L260: "spectral residuals vary from 0.34 to 0.52 % depending on the gas; it corresponds to the SNR values of 209,280, and 327" please explain (equation?) how these correspond?

L265 – 272: Earlier the authors state they have a modeled covariance. This could be a reasonable estimate and consequently the calculation could be performed and would be informative.

L293 – 297: There is no explanation of why or how the optical filter could have such an effect on the variability as it is a static or passive component. Some explanation is required to support this statement.

**Minor:**

L57 "Since Molina and Rowland (1974) have reported that CFCs accumulated in the Earth's atmosphere led to an increased rate of ozone depletion, the attention of both scientists and policymakers to the ozone hole problem has been increasing. " This statement may have been true in the 1990's but not so today.

L66 – Use of atmospheric content is not standard, often atmospheric burden when referring to the total column is used.

L73 & Amendments should be added after Montreal Protocol

L86-95 This review is not thorough. Certainly, any review of FTIR CFC efforts needs to include Rinsland 2010 and references therein.

L111: "the Tikhonov–Phillips (T–Ph) approach which is more suitable for long–lived gases with a pronounced trend." – this statement is obvious or well-known and requires a reference.

L124: "The observational system is based on a Bruker FS125HR Fourier spectrometer, but some of the equipment is non–standard." Doe the author mean in an NDACC-IRWG sense?

L125: "a non–standard spectral filter F3 was used for measurements in the spectral region with considered freons absorption bands." There is no reference for 'F3'. If a local name it should be referenced as such.

L128 – 136: The author should also refer to this as 'channeling' it's more common name and insert the in press [Blumenstock AMT 2021] for a reference.

L137: "For a preliminary assessment of the signal to noise ratio (SNR), the standard deviation (SD) of the signal", Its not clear but presumably the SD of the SNR, please clarify.

L146: "More precisely, the root mean square value (RMS) over all days for SD of the gas TCs per a day was minimized. " This not clear at all, please re-phrase.

L150: chi^2 is within the nomenclature of SFIT is a normalized part of the convergence criteria. Is it being used here in that capacity or of simply renaming RM = chi^2? This needs clarification.

Table 2: Not readable needs to be reformatted with clear rows and columns

L190: "by a priori information of the Tikhonov–Phillips" T-P is an ad hoc constraint not actually a priori information.

L192: "Unlike the OE, the T–Ph approach does not "pull" the solution to the mean profile", The OE does not pull, the retrieved profile retains the a priori value when there is no new information from the spectra. Also, not to the 'mean' rather the a priori.

L195: "the OE approach requires the use of the covariance matrices for describing the variability of the target gases profiles," not so, an array of ad hoc constrains can be applied within the OE context.

L202: (sp) choice

L209: 'and the both' is awkward maybe should be 'and both'

L211: if the author is referring to a profile scaling procedure it should be clearly stated so e.g., "first guess profile multiplier."

L216: specify section and / or page of appropriate discussion in Rodgers & Connor 2003.

L251: "geographical latitude", is redundant.

L248: does "not provide a solution" mean not converge or other issues, or both?

L310: "which does not have a systematic component during a day, to the random error." This is not clear, what is a systematic component to a random error?

EQ2 sin()

L406: 'belt' might better be 'range'

L418: "a noticeable seasonal variations" rather: "a noticeable seasonal variation"

---

## Author Response (AR2)

The authors are grateful to the reviewer for repeated careful reading of the manuscript and useful comments. We tried to resolve or eliminate controversial issues and carefully worked at the text and adding. Below are our responses to the reviewer's comments.

L34, this work does not 'control' implementation …
L38 change to …'suggests optimized'…
L48 change to …qualitatively…
L63 …transports them to…
L88 referring to a phenomenon as 'so-called' connotates that to some degree, the term is inaccurate. In this paper its simply better to be accurate. This should be re-written.
L115 'Until recently… ' is not true or clear. Please see early papers by Rinsland for data back to the 1970's
L125 the statement 'Spectral resolution of these measurements did not allow halocarbons to be measured from the surface'. Is flatly false.
L153 change informativeness to "information content"

We agree with the above comments and have corrected the text of the manuscript.

L155 while using relative transmission values may be accurate its more customary and informative to use relative absorption when discussing absorption features – this should be changed. Further these absorptions of ~ 10, 25 and 50% are not 'small'.

Indeed, such level of absorption cannot be called small, even for large zenith angles of the Sun. We have removed the mentioning of CFC-12 here, rephrased the text for CFC-11 and HCFC-22, and have also given values for solar elevation angles 50 ° - 96 and 95%, respectively.

First, the absorption of CFC-11 and HCFC-22 is not strong. Even for solar elevation of about 15°, transmission of solar radiation caused by CFC-11 absorption is greater than 90 %, for HCFC-22 is close to 75 %. For solar elevation of about 50°, these values are estimated as 96 % and 95 %, respectively.

L196 was the unsatisfactory channel modeling really due to the scatter in the retrievals?

Channeling value higher than 2% is an exceptionally large value, the usual channeling values are an order of magnitude less. These values were found in our spectra only in the first months of measurements in 2009 due to the lack of experience with such measurements. Indeed, exceptionally large channeling values increase the scatter of the results derived. We have not performed any numerical estimates and thus cannot present them. Later we improved the measurement technique so that the channeling values were usually less than 1% even with a non-standard filter. These values of channeling are successfully compensated by the SFIT4 software.

L278 "considering on the most" changed to "considering the most"
L280 "The neglecting continuum" change to "The neglection of the continuum"

We have accepted these comments and have made corrections to the text.

L165+ Major: The choice of state vector constraint is fine, the description of why and statements made to contrast with a fictitious 'OE' constraint is false and would easily mislead a reader. Further why no comparisons with a profile scaling retrieval? It's perfectly stable especially for low information content features. This point continues to be a major flaw of the text.

In the paper (Polyakov et al., 2018) we used OE and later, in the current research, we concluded that T-Ph regularization approach is preferable for using for deriving the halons from our FTIR spectra measurements. In the discussed manuscript, we did not want to raise the issue of comparing OE and T-Ph at all, which is rather complicated question and requires the special research. Therefore, we decided to exclude all questions related to this comparison from the manuscript. We are simply presenting the results derived using the T-Ph method. The profile scaling retrieval method, unfortunately, cannot be fully used within the SFIT4 shell, we explained this in the manuscript by adding the following text:

Since DOFS is close to unity for all three gases, we can consider a profile scaling approach for solving the inverse problem. However, it turned out that although the SFIT4 core solves the problem, a python script for performing the batch processing and estimating errors does not work in this case. Moreover, if profile scaling is used for all gases considered (see Table 1), the mass processing is not performed. If at least one gas (i.e., H2O) is retrieved as a profile, then mass processing is performed, but error estimates are not calculated. We compared the two approaches by analyzing all spectra measured in 2018 (measurements over 80 days) for CFC-11 retrieval. The average difference between the TCs derived by profile scaling and T-Ph approaches for this set of measurements is 0.016x10^15cm-2 or 0.33%, and SD of the difference is 0.012x10^15cm-2 or 0.26%, that is significantly less than measurement errors estimated. Therefore, to avoid problems with batch processing and error analysis, we chose the T–Ph approach.

About L: 300- Fairly Major: this discussion is mostly irrelevant. The climatology of these species is not complicated. It is reasonable enough to construct a constraint with WACCM apriori data. In fact, they are computed for SFIT and readily available.

Of course, such a construction of matrices is possible, we used it earlier. But, as we indicated above, we do not want to discuss the issues of comparison of OE and T-Ph in our manuscript, so we have removed all discussion related to the OE method.

Major: Section at 271: This discussion is interesting but inadequate. The authors only state a difference in TC from a couple tests and do not show or prove it. The spectra in the appendix do not in any way show a difference in the column amount. These slowly varying curves should have an effect on the broad region not dissimilar to the optical filter envelope – although these can change day to day. But the solar viewing instrument uses relative absorption so the absolute transmission is not in general, a concern. That is not to say the continuum has no effect, rather as the author assumes, it may well have an effect. What is required here is demonstrated proof.

We have prepared and add to the manuscript the analysis of the contribution of slowly varying absorption components, Fig. 2 and Eq. 1,2. We performed calculations for 2018 with and without taking into account the water vapor continuum, demonstrating that the magnitude of the influence of the continuum is comparable to the magnitude of seasonal variability of CFC-11

(Fig. 1), and since it is directly related to the atmospheric humidity, which has a seasonal variation, it can distort the seasonal variation of CFC-11.

This is especially true if the authors wish to make a statement that the technique be widely adopted in the NDACC as they do in the conclusions.

We have removed this statement.

This reviewer has now reviewed the document twice. The document has improved in the second version. The tables are improved the discussion of the trends is good. Still, in this second version, there are major issues the need to be corrected. There are too many minor grammar mistakes to list. The document requires a complete review as to wording and grammar. The value, efficacy and methodology of the continuum accounting needs to be described. The discussion of constraint needs to be more accurate.

We have added the text to the manuscript in accordance with the reviewer's comments, checked the text, and removed controversial topics that really should not have been discussed in this paper.